# A Pregnancy and Childhood Epigenetics Consortium (PACE) meta-analysis highlights potential relationships between birth order and neonatal blood DNA methylation

Shaobo Li[1], Natalia Spitz[2], Akram Ghantous[2], Sarina Abrishamcar[3], Brigitte Reimann[4], Irene Marques[5,6], Matt J. Silver[7], Sofía Aguilar-Lacasaña[8,9,10], Negusse Kitaba[11], Faisal I. Rezwan[11,12], Stefan Röder[13], Lea Sirignano[14], Johanna Tuhkanen[15], Giulia Mancano[16], Gemma C. Sharp[16,17], Catherine Metayer[18], Libby Morimoto[18], Dan J. Stein[19], Heather J. Zar[19,20], Rossella Alfano[4], Tim Nawrot[4], Congrong Wang[4], Eero Kajantie[21,22,23], Elina Keikkala[21,24], Sanna Mustaniemi[21,24], Justiina Ronkainen[25], Sylvain Sebert[25], Wnurinham Silva[25], Marja Vääräsmäki[21,24], Vincent W. V. Jaddoe[5,6], Robin M. Bernstein[26], Andrew M. Prentice[27], Marta Cosin-Tomas[8,9,10], Terence Dwyer[28], Siri Eldevik Håberg[29], Zdenko Herceg[2], Maria C. Magnus[29], Monica Cheng Munthe-Kaas[30], Christian M. Page[29,31], Maja Völker[14], Maria Gilles[32], Tabea Send[32], Stephanie Witt[14], Lea Zillich[14], Luigi Gagliardi[33], Lorenzo Richiardi[34], Darina Czamara[15], Katri Räikkönen[15], Lida Chatzi[35], Marina Vafeiadi[36], S. Hasan Arshad[37,38], Susan Ewart[39], Michelle Plusquin[4], Janine F. Felix[5,6], Sophie E. Moore[40], Martine Vrijheid[8,10], John W. Holloway[11,37], Wilfried Karmaus[41], Gunda Herberth[13], Ana Zenclussen[13,42], Fabian Streit[14], Jari Lahti[15], Anke Hüls[3,43], Thanh T. Hoang[44], Stephanie J. London[44] & Joseph L. Wiemels[1✉]

Higher birth order is associated with altered risk of many disease states. Changes in placentation and exposures to in utero growth factors with successive pregnancies may impact later life disease risk via persistent DNA methylation alterations. We investigated birth order with Illumina DNA methylation array data in each of 16 birth cohorts (8164 newborns) with European, African, and Latino ancestries from the Pregnancy and Childhood Epigenetics Consortium. Meta-analyzed data demonstrated systematic DNA methylation variation in 341 CpGs (FDR adjusted $P < 0.05$) and 1107 regions. Forty CpGs were located within known quantitative trait loci for gene expression traits in blood, and trait enrichment analysis suggested a strong association with immune-related, transcriptional control, and blood pressure regulation phenotypes. Decreasing fertility rates worldwide with the concomitant increased proportion of first-born children highlights a potential reflection of birth order-related epigenomic states on changing disease incidence trends.

A full list of author affiliations appears at the end of the paper.

Birth order, or the ordinal position of a child within their family, is associated with a wide variety of health outcomes. First-borns are at a higher risk for type 1 diabetes[1], high blood pressure[2], synovial sarcoma[3], metabolic diseases[4], immune diseases (including allergy[5], eczema[6], acute lymphoblastic leukemia[7,8], and lymphoma[9]). First-born children are at lower risks for other diseases including acute myeloid lymphoma[10] and non-Hodgkin lymphoma[11]. These risk associations are robust, being replicated in populations worldwide. The proportion of first-born children compared to later born children is increasing due to decreasing birth rates worldwide[12], suggest that some disease trends may be related to this changing demographic. Notably, most of the diseases listed above have exhibited increased incidence over the same time period as demographic changes leading to decreasing family size, for instance allergies and type 1 diabetes[13,14], suggesting that some proportion of the observed disease incidence trends can be attributed to this change.

Importantly, first-borns experience different gestational environments than their later-born siblings, as indexed using a variety of different biomarkers. These environments may impact later disease risk and support a biological basis for prenatal environmental conditions related to birth order. First-borns experience less sufficient placentation, higher estrogen levels, and lower insulin sensitivity, which could all contribute to subsequent post-birth disease risks[15–18]. The means and mechanisms by which these factors (related to birth order) impact childhood outcomes are not currently understood but may be crucial to efforts at understanding disease etiology and prevention. Li et al. reported that DNA methylation using a genome-wide correlation analysis of array-based DNA methylation marks of sibling pairs born after a twin birth was more correlated than sibling pairs born before a twin birth from the same mother[19]. This study suggests that DNA methylation tends to be more consistent and stabilized for later born infants, subsequent to prior deliveries; however, this study did not examine the directionality of DNA methylation alterations after twin pregnancies. In another candidate gene study[20], the DNA methylation of genes in T-cell pathways were reported to be associated with birth order, and they could in turn affect immune functions of the newborn. However, these studies had small sample sizes and could not detect DNA methylation changes with birth order on a wider genomic scale.

Here we aimed to investigate associations between neonatal DNA methylation and birth order on a genome-wide scale for the first time that we are aware of, combining results from 16 cohorts from the Pregnancy and Childhood Epigenetics Consortium (PACE). The large number of studies allowed extensive replication and consistency of findings, yielding a veritable catalog of birth order associations. Investigating differentially methylated probes (DMPs), as well as differentially methylated regions (DMRs) in infants with different birth orders may provide mechanistic insights on how birth order could impact associated developmental differences and disease risks.

## Methods

**Participating cohorts.** Sixteen cohorts from 12 countries (Germany, South Africa, Belgium, United Kingdom, Norway, Italy, Greece, Finland, Gambia, Spain, Netherlands, United States of America) were included in this study, including 8164 participants (Table 1). All studies used neonatal blood—for most this was derived from the umbilical cord, and for some from heel-prick blood spots. For a detailed description of each cohort, including DNA methylation extraction and data preprocessing steps, see Supplementary Note 1. Additional details on key birth characteristics particularly birthweight was published previously[21].

Each cohort acquired individual site-specific ethics approval as well as informed consent. The overall analysis was approved by the University of Southern California Institutional Review Board in Health Science.

**Definition of birth related variables.** Birth order refers to the number of deliveries the mother had at the time of the subject's birth. It was coded as an ordinal variable (1, 2, 3, …). Only singletons, and whose older siblings are also singletons, were included in this project, if such information is available. If multiple participants within a sample set were from the same family, only one of them was randomly included in this study to maintain independence of all study subjects. Miscarriages and abortions were not counted as delivery events. Stillbirth refers to fetal death at 20 weeks or later of pregnancy. If stillbirth information was available, it was included as a previous delivery.

**DNA methylation measurement.** Extraction of blood samples, isolation of genomic DNA, and DNA methylation array measurements were done separately by each cohort. See Supplementary Note 1 for a detailed description for each cohort. The Illumina450K array was used by 14 cohorts and the EPIC array by 8 cohorts.

**Statistics and Reproducibility.** Epigenome-wide association (EWAS) models were run in each cohort independently, with a prespecified pipeline using robust linear regression. If participants from a cohort included multiple ancestries, each ancestry was run separately. In total, there were 23 datasets, each including one ancestry from a specific cohort.

Briefly, winsorized DNA methylation beta value for each CpG was modeled as the dependent variable, with birth order (coded as 1, 2, 3,…) as a discrete independent variable. Covariates included child sex (male as 0, female as 1), technical variable to address potential batch effects, cell type proportional estimates based on the Salas et al. cord blood reference panel[22], selection factor, maternal age (years), gestational age (weeks), birthweight (gram), and maternal smoking status (nonsmoker as 0, smoker as 1). Selection factor applies when there was selection on a phenotype to create the original DNA methylation dataset for each individual study—for instance leukemia status (case/control) in the CCLS study. Note that despite the selection factor, all children were not identified as such at birth—any conditions or diseases selected were diagnosed/developed later in childhood. The main model is as follows:

$$\text{Methylation } \beta \text{ value} \sim \text{birth order (ordinal)} + \text{sex} + \text{gestational age (weeks)}$$
$$+ \text{Batch} + \text{selection factor} + \text{maternal age} + \text{birthweight}$$
$$+ \text{maternal smoking} + \text{deconvoluted cell proportion}$$

In meta-analysis, CpGs on sex chromosomes, as well as those overlapping SNPs and probes with >5% minor allele frequency in the entire population, were not included. "IlluminaHumanMethylationEPICanno.ilm10b4.hg19"[23] was used to annotate CpGs including their locations, overlapping genes or closest genes, and their regulatory regions.

Meta-analysis of all cohorts was conducted using METAL[24] weighted by inverse of standard errors, assuming fixed-effects. There were 754,340 CpGs in the final analysis that were included in at least 1 cohort. A differentially methylated probe (DMP) was defined as a CpG with false discovery rate (FDR) adjusted P value < 0.05. Heterogeneity between cohorts was measured using heterogeneity P (P_het) value output by METAL. Differentially methylation regions (DMR) were identified using "ipdmr"[25] function from the ENmix[26] R package, with default parameters. Meta-analysis and shadow meta-analysis were done in two

**Table 1 Description of participation cohorts.**

| Cohort | Country | Illumina array | Ethnicity | Sample size (n) | Mean birth weight (SD) (g) | Mean gestational age (SD) (g) | Mean maternal age (SD) (g) | Sex (n) | | Birth order (n) | |
|---|---|---|---|---|---|---|---|---|---|---|---|
| LiNA | Germany | 450 K | European | 472 | 3434.97 (470.75) | 39.78 (1.5) | 30.58 (4.52) | M | 249 | 1 | 317 |
| | | | | | | | | | | 2 | 116 |
| | | | | | | | | F | 223 | 3 | 32 |
| | | | | | | | | | | ≥4 | 7 |
| DCHS | South Africa | 450 K | Black African/ Mixed | 117 | 3101.03 (512.36) | 38.71 (1.77) | 26.61 (5.56) | M | 50 | 1 | 42 |
| | | | | | | | | | | 2 | 44 |
| | | | | | | | | F | 67 | 3 | 19 |
| | | | | | | | | | | ≥4 | 12 |
| | | EPIC | African/ Mixed | 146 | 3090.03 (559.66) | 38.99 (2.68) | 27.51 (5.95) | M | 79 | 1 | 49 |
| | | | | | | | | | | 2 | 52 |
| | | | | | | | | F | 67 | 3 | 32 |
| | | | | | | | | | | ≥4 | 13 |
| ENVIRONAGE | Belgium | 450 K | European | 182 | 3393.71 (486.92) | 39.10 (1.66) | 29.28 (4.41) | M | 96 | 1 | 101 |
| | | | | | | | | | | 2 | 81 |
| | | | | | | | | F | 86 | 3 | 0 |
| | | | | | | | | | | ≥4 | 0 |
| | | EPIC | European | 326 | 3425.00 (482.00) | 39.16 (1.63) | 30.18 (4.31) | M | 160 | 1 | 174 |
| | | | | | | | | | | 2 | 118 |
| | | | | | | | | F | 166 | 3 | 31 |
| | | | | | | | | | | ≥4 | 3 |
| POSEIDON | Germany | 450 K | European | 286 | 3415.08 (464.4) | 39.21 (1.21) | 31.58 (4.81) | M | 136 | 1 | 148 |
| | | | | | | | | | | 2 | 113 |
| | | | | | | | | F | 150 | 3 | 18 |
| | | | | | | | | | | ≥4 | 7 |
| ALSPAC | United Kingdom | 450 K | European | 744 | 3487.03 (483.68) | 39.56 (1.51) | 29.72 (4.40) | M | 363 | 1 | 354 |
| | | | | | | | | | | 2 | 278 |
| | | | | | | | | F | 381 | 3 | 85 |
| | | | | | | | | | | ≥4 | 27 |
| MoBa 1 | Norway | 450 K | European | 984 | 3645.40 (544.00) | 39.50 (1.60) | 29.90 (4.40) | M | 524 | 1 | 423 |
| | | | | | | | | | | 2 | 400 |
| | | | | | | | | F | 460 | 3 | 124 |
| | | | | | | | | | | ≥4 | 37 |
| MoBa 3 | Norway | 450 K | European | 238 | 3671.81 (555.48) | 39.65 (1.66) | 29.62 (4.41) | M | 119 | 1 | 118 |
| | | | | | | | | | | 2 | 81 |
| | | | | | | | | F | 119 | 3 | 28 |
| | | | | | | | | | | ≥4 | 11 |
| Piccolipiù | Italy | 450 K | European | 98 | 3221.07 (433.28) | 39.58 (1.57) | 33.32 (4.44) | M | 53 | 1 | 45 |
| | | | | | | | | | | 2 | 44 |
| | | | | | | | | F | 45 | 3 | 8 |
| | | | | | | | | | | ≥4 | 1 |
| Rhea | Greece | 450 K | European | 91 | 3277.80 (442.32) | 38.54 (1.30) | 29.78 (4.74) | M | 50 | 1 | 29 |
| | | | | | | | | | | 2 | 40 |
| | | | | | | | | F | 41 | 3 | 22 |
| | | | | | | | | | | ≥4 | 0 |
| PREDO | Finland | 450 K | European | 822 | 3546.00 (556.60) | 39.80 (1.60) | 33.30 (5.80) | M | 433 | 1 | 248 |
| | | | | | | | | | | 2 | 369 |
| | | | | | | | | F | 389 | 3 | 154 |
| | | | | | | | | | | ≥4 | 51 |
| | | EPIC | European | 147 | 3457.10 (517.40) | 39.80 (1.40) | 32.10 (4.90) | M | 75 | 1 | 62 |
| | | | | | | | | | | 2 | 57 |
| | | | | | | | | F | 72 | 3 | 20 |
| | | | | | | | | | | ≥4 | 8 |
| HERO-G | Gambia | EPIC | African | 115 | 3020.00 (350.00) | 39.80 (1.20) | 30.70 (7.30) | M | 68 | 1 | 12 |
| | | | | | | | | | | 2 | 21 |
| | | | | | | | | F | 47 | 3 | 13 |
| | | | | | | | | | | ≥4 | 69 |
| INMA | Spain | 450 K | European | 383 | 3271.00 (419.56) | 39.77 (1.39) | 30.36 (4.09) | M | 195 | 1 | 170 |
| | | | | | | | | | | 2 | 101 |
| | | | | | | | | F | 188 | 3 | 14 |

**Table 1 (continued)**

| Cohort | Country | Illumina array | Ethnicity | Sample size (n) | Mean birth weight (SD) (g) | Mean gestational age (SD) (g) | Mean maternal age (SD) (g) | Sex (n) | | Birth order (n) | |
|---|---|---|---|---|---|---|---|---|---|---|---|
| | | | | | | | | | | ≥4 | 0 |
| Generation R | Netherlands | 450 K | European | 1249 | 3550.00 (509.00) | 40.20 (1.50) | 31.70 (4.20) | M | 636 | 1 | 755 |
| | | | | | | | | | | 2 | 372 |
| | | | | | | | | F | 613 | 3 | 106 |
| | | | | | | | | | | ≥4 | 16 |
| Finngedi | Finland | EPIC | European | 527 | 3703.00 (470.50) | 39.93 (1.32) | 31.98 (5.28) | M | 255 | 1 | 245 |
| | | | | | | | | | | 2 | 150 |
| | | | | | | | | F | 272 | 3 | 52 |
| | | | | | | | | | | ≥4 | 80 |
| Isle of wight (IOW) | United Kingdom | 450 K | European | 104 | 3430.99 (517.40) | 39.40 (1.66) | 23.24 (2.59) | M | 46 | 1 | 66 |
| | | | | | | | | | | 2 | 25 |
| | | | | | | | | F | 58 | 3 | 11 |
| | | | | | | | | | | ≥4 | 2 |
| | | EPIC | European | 570 | 3430.00 (530.00) | 40.00 (1.50) | 27.20 (5.04) | M | 274 | 1 | 234 |
| | | | | | | | | | | 2 | 203 |
| | | | | | | | | F | 296 | 3 | 133 |
| | | | | | | | | | | ≥4 | 0 |
| California Childhood Leukemia Study (CCLS) | United States | 450 K | European | 163 | 3562.50 (581.26) | 39.21 (1.90) | 31.45 (5.75) | M | 95 | 1 | 72 |
| | | | | | | | | | | 2 | 54 |
| | | | | | | | | F | 68 | 3 | 30 |
| | | | | | | | | | | ≥4 | 7 |
| | | 450 K | Latino | 133 | 3466.80 (663.16) | 39.22 (2.40) | 27.18 (6.19) | M | 79 | 1 | 56 |
| | | | | | | | | | | 2 | 35 |
| | | | | | | | | F | 54 | 3 | 26 |
| | | | | | | | | | | ≥4 | 16 |
| | | EPIC | European | 98 | 3396.50 (554.64) | 39.30 (2.29) | 30.38 (6.24) | M | 58 | 1 | 50 |
| | | | | | | | | | | 2 | 30 |
| | | | | | | | | F | 40 | 3 | 13 |
| | | | | | | | | | | ≥4 | 5 |
| | | EPIC | Latino | 169 | 3366.44 (617.21) | 39.26 (2.18) | 26.58 (6.17) | M | 93 | 1 | 73 |
| | | | | | | | | | | 2 | 53 |
| | | | | | | | | F | 76 | 3 | 26 |
| | | | | | | | | | | ≥4 | 17 |

different institutions, one from USA (USC), the other France (IARC). Comb-p[27] was also used to identify DMRs, using meta-analyzed DMPs, to test the robustness of the "ipdmr" function.

Gene pathway enrichment analyses of DMPs were performed with "methylGSA" R package[28], using all FDR adjusted significant CpGs from the meta-analysis as inputs. Gene Ontology (GO)[29] and Kyoto Encyclopedia of Genes and Genomes (KEGG)[30] databases were both used and pathways with FDR-corrected P value < 0.05 were considered significant. Enrichment analyses of DMRs were carried out using the database for annotation, visualization and integrated discovery (DAVID)[31,32], with genes overlapping DMRs as input, focusing on GO and KEGG results.

We investigated if there was a significant increase or decrease in overlap with transcription factor (TF) binding sites among top hits. TF data for 161 transcription factors from 91 cell types were downloaded from the ENCODE project (wgEncodeRegTfbsClus-teredV3.bed). The number of CpGs among significant hits overlapping TF binding sites were compared to that of array-wide CpGs with the Fisher's exact test[33].

EWAS Open Platform[34] was used to conduct trait enrichment analysis, in order to identify if significant CpGs from our study were associated with other phenotypes included in EWAS Atlas[35]. Associations of DNA methylation levels in the blood and brain were inferred using BECon[36]. Expression levels of genes related to DMPs in different tissues were queried on the GTEx portal[37].

Finally, the associations between methylation levels of significant CpGs and expressions of nearby genes (cis-expression quantitative trait methylation, cis-eQTMs) in the blood were queried from published results[38] from the Human Early Life Exposome (HELIX) project[39].

**Sensitivity analyses.** We conducted several sensitivity analyses to test the robustness of our results.

For the top 20 CpGs from the meta-analysis, we conducted leave-one-out (LOO) analyses, excluding one cohort at a time, to observe if the results were driven by one specific cohort. Forest plots showing LOO results were plotted with 'forestplot' function in the 'forestplot' R package[40].

To investigate whether the associations between DNA methylation and birth order were different in different ancestries, we repeated the meta-analyses in European participants ($n = 7484$) and African participants ($n = 378$) separately. Ancestry specific analysis was not run in Latinos because Latino participants had the smallest sample size, and they all came from one cohort.

In addition, miscarriages and abortions have arguably smaller physiological effects than full term pregnancy; however, their effects on neonatal DNA methylation of future babies are unclear. Therefore, for cohorts with miscarriage/abortion information available, we carried out sensitivity analysis counting miscarriages and abortions as a delivery.

Lastly, maternal weight gain was reported to be associated with placental DNA methylation alterations[41], which in turn could affect neonatal methylation. To test this, we adjusted for maternal weight gain as an additional sensitivity analysis in cohorts with this information.

**Reporting summary**. Further information on research design is available in the Nature Portfolio Reporting Summary linked to this article.

## Results

**Meta-analysis identified significant CpG probes associated with birth order**. Our meta-analysis included all 23 datasets from 16 cohorts identified 341 CpGs differentially methylated at FDR adjusted $P$ value < 0.05 (Fig. 1, Supplementary Data 1). In these and all data presented, positive coefficients refer to higher (hyper-) DNA methylation with later birth order compared to earlier, and negative coefficients refer to lower (hypo-) methylation with later birth order compared to earlier. The most significant CpG (cg09249800, FDR adjusted P value = $7.24 \times 10^{-6}$) was in a CpG island in the gene body of *ACOT7*. The second most significant CpG was located in the transcription start site (TSS) of *LOC650226*, located in a Chromosome 7 peak overlapping shore and island regions of *LOC650226* and *ZNF727* genes (Fig. 1, Supplementary Data 2). CpG sites in the promoter regions of *FAM169A* (cg04654716, FDR adjusted $P$ value = $4.40 \times 10^{-4}$) and *LIF* (cg19539004, FDR adjusted $P$ value = $4.40 \times 10^{-4}$) were also among the top hits. See Table 2 for annotation of all significant hits including their genomic coordinates, meta-analysis $I^2$ value and additional outputs. We also computed the top associations with a statistical model examining first birth versus all subsequent births as a group (bivariate analysis) (Supplementary Data 3).

A total of 1 KEGG and 43 GO pathways were enriched among these 341 DMPs (Supplementary Data 4), including those involved in cell growth development (germ cell development, multicellular organism reproduction, growth factor activity etc.) and leukocyte activation and migration (leukocyte transendothelial migration, positive regulation of B cell activation, regulation of leukocyte chemotaxis etc.).

We collected data on all 161 transcription factors (TFs) from ENCODE ChiP-seq database and tested if birth order related CpGs were more or less likely to overlap with TF bindings sites. As a result, 10 TF binding sites (MAZ, CTCF, POLR2A, RAD21, EZH2, ZBTB7A, GATA3, GATA2, TAL1, POU5F1) were enriched, while 13 (ATF1, CREB1, NFYA, GTF2F1, CEBPD, ELK1, RFX5, TAF7, RELA, KDM5B, E2F4, PML, SIN3AK20)

were depleted (i.e., significantly under-represented) among these CpGs.

Trait enrichment analysis suggested that birth order-associated hits were also associated with 69 other traits (Supplementary Data 5), the top 4 of which were all immune-related phenotypes including allergic sensitization ($P$ value = $5.90 \times 10^{-96}$), fractional exhaled nitric oxide ($P$ value = $1.53 \times 10^{-69}$), childhood asthma ($P$ value = $3.42 \times 10^{-60}$), and atopy ($P$ value = $3.42 \times 10^{-60}$). Smoking ($P$ value = $1.98 \times 10^{-38}$), maternal smoking ($P$ value = $5.28 \times 10^{-23}$), down syndrome ($P$ value = $2.58 \times 10^{-20}$) and neurodevelopmental presentations and congenital anomalies (ND/Cas) ($P$ value = $8.31 \times 10^{-18}$) were also among top enriched traits.

Forty out of the 341 significant CpGs (11.73%) were previously reported to be cis-eQTMs in blood (Supplementary Data 6), with some of the CpGs associated with multiple transcripts. This proportion was much higher than that for all CpGs on the 450 K array (2.37%). For example, the methylation level of cg04654716 was reported to be positively associated with *FAM169A* expression level (eQTM $P$ value = $6.24 \times 10^{-7}$).

DNA methylation levels are often tissue-specific, and because we analyzed DNA from blood, we wanted to evaluate whether we could infer DNA methylation levels of these 341 birth-order related CpGs in the brain, because trait hits above seemed to be very relevant to neural functions. By querying published dataset by Edgar et al.[36] which reported concordance of DNA methylation in the blood and the brain, 277 birth order related CpGs had blood-brain association data available (Supplementary Data 7), and 113 (40.79%) CpGs among them had an absolute Spearman correlation coefficient bigger than 0.2. Interestingly most of the genes we mentioned as top birth order-associated hits exhibited enhanced gene expression in brain tissues compared with other tissues (including *PRRT1, PLEKHB1, ACOT7, FAM169A, ZBED9*) (Supplementary Fig. 1–5).

**Differentially methylated regions associated with birth order**. We identified 1,107 DMRs associated with birth order (Table 3, Supplementary Data 8). Functional annotations with genes overlapping these DMRs by DAVID[31] identified 17 significant pathways (adjusted $P$ value < 0.05). Eleven (64.70%) of them are related to DNA transcription regulation, 3 of them likely related

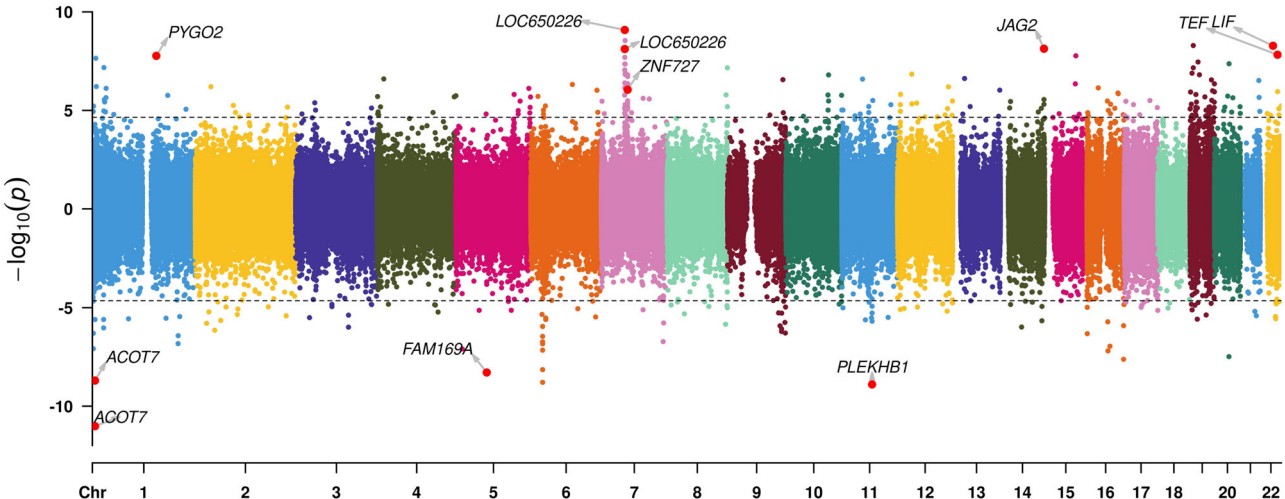

**Fig. 1 Bi-direction Miami plot showing associations between DNA methylation and birth order.** Bi-directional Miami plot showing the results of meta-analysis of the association between DNA methylation and birth order, adjusting for sex, maternal age, gestational age, birthweight, maternal smoking, batch effects, selection factor, and cell proportions. Directions of the associations were shown on the Y-axis, with positive associations above Y = 0 and negative associations below. Threshold of significance after false-discovery rate (FDR) correction is shown in a dashed horizon line. A total of 341 CpGs were significant after FDR multiple correction, the threshold of which was shown with a dashed line.

**Table 2 Top 25 significant CpGs associated with ascending birth order from meta-analysis.**

| cpg | Effect | Raw P value | FDR adjusted P value | Coefficient direction | P het[a] | Relation to Island | UCSC RefGene Name | UCSC RefGene Group |
|---|---|---|---|---|---|---|---|---|
| cg09249800 | −0.0023 | 9.60E−12 | 7.24E−06 | (see figure) | 0.002 | Island | ACOT7 | Body |
| cg26262573 | 0.0078 | 8.31E−10 | 3.00E−04 | (see figure) | 0.005 | S_Shore | LOC650226 | TSS200 |
| cg26776957 | −0.0028 | 1.27E−09 | 3.00E−04 | (see figure) | 0.841 | OpenSea | PLEKHB1;PLEKHB1 | TSS200;TSS1500 |
| cg26865747 | −0.0105 | 1.62E−09 | 3.00E−04 | (see figure) | 0.003 | N_Shore |  |  |
| cg11699125 | −0.0023 | 1.99E−09 | 3.00E−04 | (see figure) | 0.000 | Island | ACOT7 | Body |
| cg03779117 | 0.0034 | 2.86E−09 | 3.59E−04 | (see figure) | 0.926 | Island |  |  |
| cg09377704 | 0.0025 | 5.11E−09 | 4.40E−04 | (see figure) | 0.579 | N_Shore |  |  |
| cg04654716 | −0.0023 | 5.12E−09 | 4.40E−04 | (see figure) | 0.113 | S_Shore | FAM169A | TSS1500 |
| cg19539004 | 0.0017 | 5.25E−09 | 4.40E−04 | (see figure) | 0.589 | S_Shore | LIF | TSS1500 |
| cg12763978 | −0.0064 | 7.13E−09 | 4.82E−04 | (see figure) | 0.018 | N_Shore |  |  |
| cg05335315 | 0.0011 | 7.43E−09 | 4.82E−04 | (see figure) | 0.647 | OpenSea | JAG2 | TSS1500 |
| cg17328716 | 0.0056 | 7.67E−09 | 4.82E−04 | (see figure) | 0.062 | S_Shore | LOC650226 | TSS200 |
| cg20534570 | 0.0023 | 1.49E−09 | 8.57E−04 | (see figure) | 0.048 | OpenSea | TEF | TSS1500 |
| cg25738326 | 0.0015 | 1.70E−08 | 8.57E−04 | (see figure) | 0.035 | OpenSea |  |  |
| cg22608655 | 0.0021 | 1.70E−08 | 8.57E−04 | (see figure) | 0.180 | N_Shore | PYGO2 | Body |
| cg15730180 | 0.004 | 2.03E−08 | 9.58E−04 | (see figure) | 0.552 | N_Shore |  |  |
| cg20014974 | 0.0026 | 2.26E−08 | 1.00E−03 | (see figure) | 0.761 | N_Shore |  |  |
| cg26443093 | −0.0012 | 2.40E−08 | 1.00E−03 | (see figure) | 0.662 | N_Shore | ZFPM1 | Body |
| cg01649647 | −0.0013 | 3.28E−08 | 1.30E−03 | (see figure) | 0.185 | OpenSea | MYL9 | Body |
| cg00741634 | 0.0021 | 3.50E−08 | 1.32E−03 | (see figure) | 0.599 | S_Shore | KIAA0892;SF4 | Body; TSS1500 |
| cg17675882 | 8.00E−04 | 4.29E−08 | 1.46E−03 | (see figure) | 0.041 | N_Shore | NDRG3 | 5'UTR |
| cg03122674 | 0.0068 | 4.42E−08 | 1.46E−03 | (see figure) | 0.011 | Island | LOC650226 | Body |
| cg15134787 | 0.0062 | 4.45E−08 | 1.46E−03 | (see figure) | 0.084 | N_Shore |  |  |
| cg02119982 | −3.00E−04 | 6.41E−08 | 1.88E−03 | (see figure) | 0.417 | OpenSea | NOD2 | Body |
| cg22019158 | 0.0015 | 6.70E−08 | 1.88E−03 | (see figure) | 0.185 | N_Shelf | AHDC1 | 5'UTR |

[a]P het: Heterogeneity P values output by METAL
[b]The direction of association between later birth order and DNA methylation is positive (+) or negative (−) in each cohort. Cohorts are ordered from the largest to the smallest. If the CpG was not assessed in a cohort (due to its being missed on the array) a "?" will be displayed.

to transcription regulation (17.65%), and only 2 (GO:0098978 glutamatergic synapse, and GO:0005887 integral component of plasma membrane) (11.76%) are not related to this function (Supplementary Data 9).

**Sensitivity analyses**. We ran several sensitivity analyses to test the robustness of our results. We did leave-one-out analyses for top 20 hits from our analysis, each time excluding one dataset from the meta-analysis, to test if results were heavily influenced by any one dataset. For all top 20 CpGs, leaving datasets out one by one did not change the significance of our results. Effect sizes were all in the same direction as the main model, and none of the 95% confidence interval (CI) of the meta-analysis estimates crossed zero (Supplementary Figs. 6–9).

Since our participants were from different ancestral groups but predominantly European, we conducted meta-analyses in European ($n = 7484$) and African ($n = 378$) ancestries separately to observe ancestry-specific birth order related CpGs (Supplementary Data 1, Supplementary Data 10). In European participants, there were 316 significant CpGs after multiple correction, while in African participants alone, only 1 CpG remained significant (Supplementary Data 2), likely due to small sample size. 117 of the 341 significant CpGs from the main model were also significant In European participants, and all CpGs had the same direction of effects (Supplementary Data 1). However, in participants of African ancestry, 273 CpGs out of the 341 CpGs (80.06%) were in the same direction as the main model, and none of these 341 CpGs were significant in African participants alone (Supplementary Data 1).

We also controlled for maternal weight gain as an additional variable, and results were highly consistent, including the 341 significant CpGs from the original model (Fig. 2a). We also counted abortions/ miscarriages as a birth event, and results were similar to our main models (Fig. 2b).

## Discussion

In this study, we combined multiple cohorts from 12 countries, including participants of European, African and Latino self-reported ancestries, and identified 341 CpGs whose DNA methylation levels were associated with birth order. This was the first multi-cohort large-scale EWAS study investigating the associations between neonatal DNA methylation and birth order. As no single cohort was specifically designed to examine DNA methylation and birth order our results may be considered exploratory, however the strength of the PACE Consortium allows confirmatory replication and validation.

Birth order has been associated with multiple diseases and does not have a genetic cause. Therefore, it is of interest to investigate whether epigenetic alterations, especially DNA methylation observable at birth, is associated with birth order. These epigenetic alterations may mediate the impact of birth order on disease risk, and can serve as a roadmap of candidate biomarkers to investigate such risk. To establish a robust set of birth order-associated biomarkers, we conducted an EWAS meta-analysis including multiple datasets from cohorts around the world. We found numerous CpGs differentially methylated in relation to birth order, with some associated with gene expression in tissues that have birth order disease associations such as the brain, immune system, and cardiovascular system. The dramatic fall in fertility rates worldwide over the preceding decades and projections for the future are leading to a higher proportion of first-born individuals with certain future continuation of such trends; in addition the contribution of variance in DNA methylation impacted by birth order and its associated diseases is of strong interest to the Developmental Origin of Health and Disease (DOHaD) community.

The most significant CpG was cg09249800 (adjusted $P$ value $7.24 \times 10^{-6}$), in the gene body region of *ACOT7*. The encoded protein hydrolyzes palmitoyl-coenzyme A (palmitoyl-CoA), and was reported to be associated with mesial temporal lobe epilepsy[42]. Interestingly, a previous GWAS study[43] identified a SNP (rs11121611) within *ACOT7* to be associated with "asthma exacerbation measurement, response to corticosteroid". However, cg09249800 (chr1:6341287, Hg19) was about 25 kb upstream of rs11121611 (chr1:6367119, Hg19), and it was not reported to be a cis-eQTM of *ACOT7* (Supplementary Data 6).

While cg09249800 was the most statistically significant association, its effect size ($-2.8 \times 10^{-3}$) was nearly 4 times smaller than strongest effect size CpG which was cg26865747, (coefficient = 0.0105), proximal to the *SCAND3* gene, a zinc finger transcription factor implicated in tumor proliferation and invasion[44]. Significant individual CpG sites ranged in effect sizes from 0.0001 to 0.01, namely over two orders of magnitude, and 70 of the 341 CpGs had larger effect sizes than the most significant single CpG site at *ACOT7*. Other significant hits were of interest. For example, there was a prominent cluster of 10 CpGs overlapping the *LOC650226* and *ZNF727* genes. In addition, all significant CpGs overlapping *ZNF727* were reported to be cis-eQTMs, meaning their DNA methylation levels were associated with expression levels of the *ZNF727* gene. The reasons for their associations with birth order requires further investigation, although it is interesting that all 10 CpGs were also reported by Håberg et al.[45] to have a significantly lower DNA methylation level in babies born with assisted reproductive technology (ART) compared to naturally conceived babies (FDR adjusted $P$ value $< 9.86 \times 10^{-5}$). Interestingly in our study, later-borns were more methylated than first-borns in this region. It was not clear why DNA methylation patterns vary in this manner. A potential explanation is that the later order a child was born, the more established the pregnancy process becomes including placentation, leading to more stable nutrition status promoting physiologic homeostatic DNA methylation patterns. We did not evaluate the relationship of *ZNF727* DNA methylation to postnatal outcomes, but such an effort would be valuable, particularly in the modern era as family size in some countries has decreased compared to historical trends. Either way, additional data is needed to elucidate answers to whether DNA methylation alterations may mediate some of the disease associations ascribed to birth order.

The CpG site cg04654716 (effect size $-0.0023$, adjusted $P$ value $5.12 \times 10^{-9}$) in the transcription start site of *FAM169A* is also of interest. Similar to CpGs overlapping *ZNF727*, cg04654716 was also reported to be a cis-eQTM, whose DNA methylation level was positively related to *NSA2* expression level. SNPs in *NSA2* were also associated with metabolism-related traits in many GWAS studies (low density lipoprotein cholesterol measurement[46,47], total cholesterol measurement[46,47], linoleic acid measurement[48], omega-6 polyunsaturated fatty acid measurement[48,49], and HMG CoA reductase inhibitor use measurement[50]). Interestingly, metabolic function was also related to birth order[4], and its causal pathway is worth further investigation.

Most of the participants in this study were of European ancestry, and unsurprisingly, in European participants alone, the effect sizes of all significant CpGS were in the same directions as the main model, while in African participants, about 80% significant CpGs were in the same direction (Supplementary Data 2). Further investigation on how these CpGs were related to birth order in other ancestries including Asians and Latinos is required.

**Table 3 Top 10 DMRs associated with birth order from meta-analysis.**

| chr | Start | End | Width | P | FDR adjusted P value | # of probes | Gene symbol | Distance2TSS |
|---|---|---|---|---|---|---|---|---|
| chr6 | 28601270 | 28602543 | 1274 | 3.70E−36 | 4.17E−33 | 11 | ZBED9 | −46158 |
| chr7 | 56514964 | 56516425 | 1462 | 9.68E−33 | 5.46E−30 | 11 | LOC650226 | 0 |
| chr11 | 73357018 | 73357612 | 595 | 9.44E−25 | 3.55E−22 | 9 | PLEKHB1 | 0 |
| chr6 | 32120202 | 32121261 | 1060 | 3.25E−22 | 9.17E−20 | 26 | PRRT1 | 0 |
| chr7 | 63360692 | 63361617 | 926 | 2.13E−20 | 4.79E−18 | 6 | LINC01005 | 128863 |
| chr7 | 63385989 | 63387147 | 1159 | 3.36E−19 | 5.60E−17 | 8 | LINC01005 | 103333 |
| chr7 | 57483672 | 57484819 | 1148 | 3.48E−19 | 5.60E−17 | 6 | MIR3147 | 10941 |
| chr7 | 63505637 | 63506148 | 512 | 4.39E−19 | 6.18E−17 | 5 | ZNF727 | 0 |
| chr7 | 57471758 | 57473294 | 1537 | 6.07E−19 | 7.60E−17 | 6 | MIR3147 | 0 |
| chr9 | 131154346 | 131156014 | 1669 | 5.21E−18 | 5.87E−16 | 7 | MIR219B | 0 |

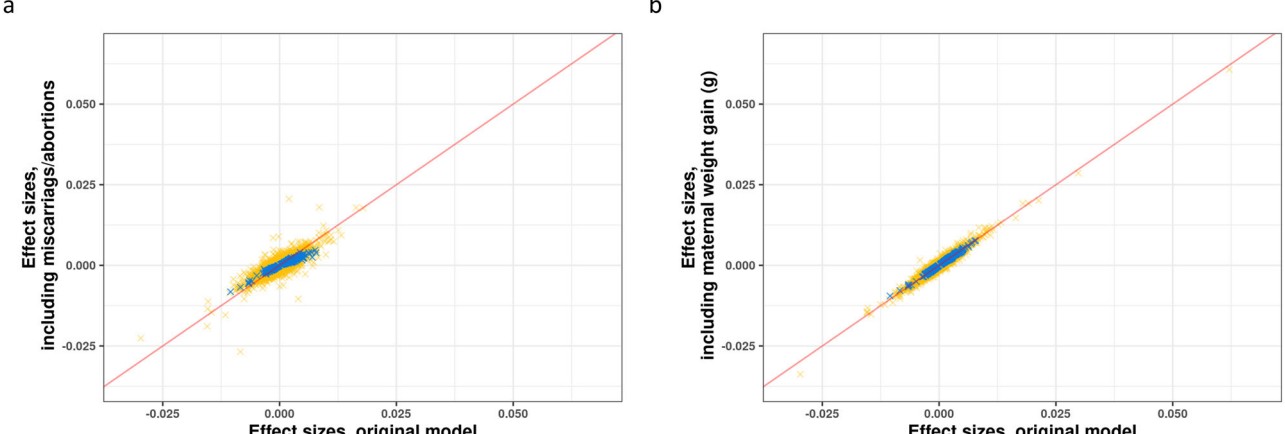

**Fig. 2 Comparisons of results from sensitivity models to the main model. a** Effect sizes from the main models (*X*-axis) plotted against effect sizes from models including maternal weight gain as an additional covariate (*Y*-axis). *Y* = *X* line was plotted in a red line. The 341 significant CpGs from the main models were colored blue, and all other non-significant CpGs were colored yellow. **b** Effects sizes from the main models (*X*-axis) plotted against effect sizes from models counting miscarriage/abortion as a birth event (*Y*-axis), similar to that of (**a**).

In trait enrichment analyses, birth order related CpGs were also associated with 69 traits (Supplementary Data 5), especially allergy-related features including allergic sensitization, childhood asthma, atopy, serum immunoglobulin E levels, allergic asthma, wheeze, respiratory allergies, primary Sjögren's Syndrome, systemic lupus erythematosus, multiple sclerosis and cow's milk allergy. Interestingly, first-borns were also reported to have a higher risk of allergy[51] and eczema was previously reported to be associated with birth order[6,51]. Other immune related features were also significantly associated with birth order, for example, psoriasis, B acute lymphoblastic leukemia (B-ALL) and acute chorioamnionitis (aCA). Among those, B-ALL was reported to be more common in first-borns[7].

Several other traits previously reported to be related to birth order were also identified, including blood pressure[2] (cardiac autonomic responses (deceleration capacity), diastolic blood pressure, systolic blood pressure, atherosclerosis, preeclampsia, maternal hypertensive disorders in pregnancy), metabolism[4] (serum liver enzyme levels (alanine aminotransferase, ALT), serum liver enzyme levels (gamma-glutamyl transferase, GGT), metabolic trait, hepatic steatosis, hepatic fat), birth weight[52] and body mass index (BMI)[53]. The top association in our DMR analysis was *ZBED9*, recently identified as a regulatory gene for blood pressure[54].

Additional traits were associated with birth order related CpGs in our study, but have not previously reported to be associated with birth order. These include abnormal karyotype related traits (Down syndrome, Klinefelter syndrome), and several neural function-related traits (soluble tumor necrosis factor receptor 2 levels in plasma, neurodevelopmental presentations and

congenital anomalies (ND/CAs), schizophrenia, myalgic encephalomyelitis/chronic fatigue syndrome, leukoaraiosis). More investigation could reveal whether these traits are also related to birth order, and how birth order related epigenetic changes might contribute to such relationships.

Neurological traits previously assessed in relationship to birth order include intelligence, in which first-borns tended to display higher levels[55–57]. As our samples were collected from blood, these enriched neural related traits also led us to investigate the consistency of methylation levels of birth order associated CpGs in blood and brain. Of all the significant CpGs whose blood-brain association data were available, 40.79% had modest to strong associations (absolute Spearman correlation coefficient >0.2). When we confined our trait enrichment analysis to CpGs whose methylation levels were highly correlated in blood and brain only, similar traits were enriched, and neurodevelopmental presentations and congenital anomalies (ND/CAs) became the second most enriched phenotype (Supplementary Data 5).

We identified 1,107 DMRs associated with birth order. Enrichment analysis of these genes showed that almost all significant pathways were related to regulation of gene transcription. More work is needed to understand what proteins were regulated and in which direction to elucidate more specific information that could impact human development. Only 2 enriched pathways were not transcription related, one of them being the glutamatergic synapse pathway. Glutamatergic synapse is involved in neural network development, and is essential for transferring and processing information[58], which may contribute to the association between birth order and intelligence.

There were some shortcomings with this study. While investigating birth order, we examined unrelated individuals, instead of same-family siblings. This inevitably introduced noise and decreased the reliability of our findings due to genetic, socio-economic status, and cultural differences between study participants both within and across study cohorts. The choice to remove any genetically related study subjects (by family relationships) maintains the independence of every study subject, randomizing unmeasured confounders whereas we statistically controlled for key known confounders—including sex, cell type distributions, maternal age, gestational age, birthweight, and maternal tobacco use (when available). We were not able to control for socio-economic status characteristics (such as maternal education or family income), characteristics which will impact disease risk and potentially DNA methylation profiles. Moreover, our study was designed to identify associations only, and is not capable of demonstrating causality or mediation by DNA methylation and birth order-related diseases. Instead, the data provides a catalog of candidates for future research. Also, despite the worldwide scope and large number of studies in the PACE Consortium, we lack extensive data on some race/ethnic groups particularly non-Europeans including Latinos, Africans and Asians. It would be of interest to investigate how birth order is associated with DNA methylation changes in these groups, and how they vary from the conclusions in this study. Another weakness is the variability of average family size among the various PACE consortium studies (which vary with regards to fertility rates) which might affect power to detect effects from higher birth orders. Also, our focus on neonatal blood exclusively may limit the discovery aspects of our data for other tissues, along with the limited coverage of the epigenome afforded by the Illumina array platforms used. Strengths of this study include the large sample size afforded by the PACE on a common analysis platform, and the presumed consistency in our main predictor variable (birth order) which should have a universal description worldwide and similar physiologic impacts across countries. The use of cord blood is also a major strength in that a molecular phenotype was captured before onset of many of the associated traits. The consistency of our top CpG hits across cohorts argues for true and meaningful associations which may prove to have further resources for the maternal/fetal health research community.

We note that first-borns, compared to their later-born siblings, have a variety of postnatal environmental differences which may also impact disease risk. First-borns are typically exposed to infectious agents later in their childhood development; indeed, birth order was often used as a proxy for infection timing[59–61]. Such postnatal experiences including child rearing practices and postnatal infections are commonly conjectured to be mediators for birth order's health and disease impacts. These factors are not likely to be related to pre-birth environments and are not a subject of the current analysis but are important postnatal mediators. Future studies should robustly evaluate both the prenatal and postnatal mediators of birth order on disease risk —including DNA methylation at birth.

In conclusion, our results from multiple datasets showed with high confidence that birth order has a widespread and consistent association with DNA methylation in the cord blood of newborns. These differences provide a catalog of associations which can be assessed as causal mediators in the etiology of health conditions related to birth order.

## Data availability

Blood samples and raw genetic data of neonatal subjects from each cohort are governed by their respective institutions and/or government agencies, and mostly could not be shared publicly without specific approvals. For example, for data from first author cohort, California Childhood Leukemia Study (CCLS), we respectfully are unable to share raw, individual genetic data freely with other investigators. Should we be contacted by other investigators who would like to use the data; we will direct them to the California Department of Public Health Institutional Review Board to establish their own approved protocol to utilize the data, which can then be shared peer-to-peer.

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

## Acknowledgements

We thank all the participants and their families for this study. We also acknowledge all funding agencies of cohorts involved in this study, as mentioned in the supplementary methods. The overall study at University of Southern California was funded by the National Institute of Environmental Health Sciences (P01ES018172, R01ES09137, R01CA185058), the National Cancer Institute (R01CA175737), and the US Environmental Protection Agency (RD83451101, RD83615901).

## Author contributions

Conceptualization: J.L.W., S.J.L., S.L., A.G.; Meta-analysis and Shadow analysis: S.L., N.S.; Cohort-specific analysis: S.L., N.S., A.G., S.A., B.R., I.M., M.J.S., S.A.-L., N.K., F.I.R., S.R., L.S., J.T., G.M., G.C.S., C.M., L.M., D.J.S., H.J.Z., R.A., T.N., C.W., E. Kajantie, E. Keikkala, S.M., J.R., S.S., W.S., M.V., V.W.V.J., R.M.B., A.M.P., M.C.-T., T.D., S.E.H., Z.H., M.C.M., M.C.M.-K., C.M.P., M.V., M.G., T.S., S.W., L.Z., L.G., L.R., D.C., K.R., L.C., M.V., S.H.A., S.E., M.P., J.F.F., S.E.M., M.V., J.W.H., W.K., G.H., A.Z., F.S., J.L., A.H., T.T.H., S.J.L., J.L.W.; Writing—Original Draft: S.L., J.L.W.; Writing—Review and Editing: S.L., N.S., A.G., S.A., B.R., I.M., M.J.S., S.A.-L., N.K., F.I.R., S.R., L.S., J.T., G.M., G.C.S., C.M., L.M., D.J.S., H.J.Z., R.A., T.N., C.W., E. Kajantie, E. Keikkala, S.M., J.R., S.S., W.S., M.V., V.W.V.J., R.M.B., A.M.P., M.C.-T., T.D., S.E.H., Z.H., M.C.M., M.C.M.-K., C.M.P., M.V., M.G., T.S., S.W., L.Z., L.G., L.R., D.C., K.R., L.C., M.V., S.H.A., S.E., M.P., J.F.F., S.E.M., M.V., J.W.H., W.K., G.H., A.Z., F.S., J.L., A.H., T.T.H., S.J.L., J.L.W.; Visualization: S.L.; Project administration: J.W., S.L.

## Competing interests

The authors declare no competing interests. Where authors are identified as personnel of the International Agency for Research on Cancer/ World Health Organization, the authors alone are responsible for the views expressed in this article and they do not necessarily represent the decisions, policy or views of the International Agency for Research on Cancer/ World Health Organization.

## Additional information

[1]Center for Genetic Epidemiology, Department of Population and Public Health Sciences, University of Southern California, Los Angeles, California, USA. [2]Epigenomics and Mechanisms Branch, International Agency for Research on Cancer, Lyon, France. [3]Department of Epidemiology, Rollins School of Public Health, Emory University, Atlanta, GA, USA. [4]Centre for Environmental Sciences, UHasselt, Agoralaan, Building D, 3590 Diepenbeek, Belgium. [5]The Generation R Study Group, Erasmus MC, University Medical Center Rotterdam, Rotterdam, the Netherlands. [6]Department of Pediatrics, Erasmus MC, University Medical Center Rotterdam, Rotterdam, the Netherlands. [7]Medical Research Council Unit The Gambia at the London School of Hygiene and Tropical Medicine, London, UK. [8]ISGlobal, Institute for Global Health, Barcelona, Spain. [9]Universitat Pompeu Fabra (UPF), Barcelona, Spain. [10]CIBER Epidemiología y Salud Pública, Madrid, Spain. [11]Human Development and Health, Faculty of Medicine, Southampton General Hospital, University of Southampton, Southampton, UK. [12]Department of Computer Science, Aberystwyth University, Aberystwyth, Ceredigion SY23 3DB, UK. [13]Department of Environmental Immunology, Helmholtz Centre for Environmental Research –UFZ, Leipzig, Germany. [14]Department of Genetic Epidemiology in Psychiatry, Central Institute of Mental Health, Medical Faculty Mannheim, University of Heidelberg, Mannheim, Germany. [15]Department of Psychology and Logopedics, University of Helsinki, Helsinki, Finland. [16]MRC Integrative Epidemiology Unit, Population Health Sciences, Bristol Medical School, University of Bristol, Bristol, UK. [17]School of Psychology, Faculty of Health and Life Sciences, University of Exeter, Exeter, UK. [18]School of Public Health, University of California Berkeley, Berkeley, California, USA. [19]SAMRC Unit on Risk & Resilience in Mental Disorders, Dept of Psychiatry & Neuroscience Institute, University of Cape Town, Rondebosch, South Africa. [20]Department of Paediatrics and Child Health, Red Cross War Memorial Children's Hospital, University of Cape Town, Rondebosch, South Africa. [21]Clinical Medicine Research Unit, Medical Research Center Oulu, Oulu University, Hospital and University of Oulu, Oulu, Finland. [22]Department of Clinical and Molecular Medicine, Norwegian University of Science and Technology, Trondheim, Norway. [23]Pediatric Research Centre, Children's Hospital, Helsinki University Hospital and University of Helsinki, Helsinki, Finland. [24]Population Health Unit, Department of Public Health and Welfare, Finnish Institute for Health and Welfare, Oulu, Finland. [25]Research Unit of Population Health, Faculty of Medicine, University of Oulu, Oulu, Finland. [26]Department of Anthropology and Institute of Behavioral Science, University of Colorado Boulder, Boulder, CO, USA. [27]MRC Unit The Gambia at the London School of Hygiene & Tropical Medicine, Fajara, The Gambia. [28]Nuffield Department of Women's & Reproductive Health, University of Oxford, John Radcliffe Hospital, Oxford OX3 9DU, UK. [29]Centre for Fertility and Health, Norwegian Institute of Public Health, Oslo, Norway. [30]Department of Pediatric Oncology and Hematology, Oslo University Hospital, Norwegian Institute of Public Health, Oslo, Norway. [31]Department of Physical Health and Aging, Division for Mental and Physical Health, Norwegian Institute of Public Health, Oslo, Norway. [32]Department of Psychiatry and Psychotherapy, Central Institute of Mental Health, Medical Faculty Mannheim, University of Heidelberg, Mannheim, Germany. [33]Woman and Child Health Department, Ospedale Versilia, AUSL Toscana Nord Ovest, Pisa, Italy. [34]Department of Medical Sciences, University of Turin, CPO Piemonte, Turin, Italy. [35]Department of Population and Public Health Sciences, Keck School of Medicine of USC. University of Southern California, Los Angeles, CA, USA. [36]Department of Social Medicine, Faculty of Medicine, University of Crete, Heraklion, Greece. [37]Clinical and Experimental Sciences, Faculty of Medicine, University of Southampton, Southampton, UK. [38]David Hide Asthma and Allergy Research Centre, Isle of Wight, UK. [39]College of Veterinary Medicine, Michigan State University, East Lansing, MI, USA. [40]Department of Women & Children's Health, King's College London, London, UK. [41]Division of Epidemiology, Biostatistics, and Environmental Health, University of Memphis, Memphis, TN, USA. [42]Perinatal Immunology, Medical Faculty, Saxonian Incubator for Clinical Translation (SIKT), University of Leipzig, Leipzig, Germany. [43]Gangarosa Department of Environmental Health, Rollins School of Public Health, Emory University, Atlanta, GA, USA. [44]Epidemiology Branch, National Institute of Environmental Health Sciences, National Institutes of Health, Research Triangle Park, NC, USA. ✉email: wiemels@usc.edu

