## [Peer Review File · Communications Biology]

Reviewers' comments:

Reviewer #1 (Remarks to the Author):

Thank you for the opportunity to review this interesting manuscript. The authors report the results of a multi-cohort meta-analysis showing over 300 CpG sites that are differentially methylated with regards to child birth order. I really enjoyed the clear presentation of (very complex) findings and the analytic choices and sensitivity analyses seemed carefully thought through and appropriate. I have a number of suggestions for strengthening the manuscript.

In the introduction, I thought too much space was dedicated to talking about postnatal explanations for birth-order differences, when this paper is focusing on prenatal explanations. The second half of the first paragraph could be shortened, and more specific information could be provided in the second paragraph, to give more information about potential prenatal processes linking birth order to differential disease risk.

I also thought the study by Li et al about sibling pairs and twin births (line 141) could be better explained. Which siblings' DNA methylation were being compared? One of the twins and earlier/later-born siblings?

The authors state that some studies included multiple children from the same family. How common was this? If there are enough cases, it would strengthen the authors' findings to investigate DNA methylation of birth-order related CpGs WITHIN families to see if the direction of associations found in between-family analyses plays out within-families. This was a limitation raised in the discussion. If the data do not exist to be able to address within-family differences here, I think more should be said about this in the limitations. We know that family size is not a randomly distributed characteristic – it likely relates to parent and family characteristics that are also likely to be associated with disease risk (e.g., parental education).

How prevalent were stillbirths in the sample? There are likely different physical effects associated with stillbirth vs live birth. Were sensitivity analyses conducted with and without these contributing to birth order (similar to how this was done for miscarriages and abortions?)

How was participant ancestry determined? Were there multiple ancestry children/families?

Can the authors clarify what the 'selection factor' (line 186) refers to?

Birth order was treated as an ordinal variable in analyses. I think the results then reflect the assumption that each additional birth has an additive impact on DNA methylation. But, is there any evidence that there could be threshold effects? That is, are first-borns systematically different from later-borns, but later-borns do not differ from one another? Or, there is a difference up to 3 births and then no further differences? Could this be explored in these data?

It was unclear from the description of analysis steps whether the DMR analysis was only conducted on the meta-analytic sample, or whether it was conducted separately by cohort and then meta-analyzed (as the DMPs were). Please clarify.

Were other pregnancy complications investigated? The authors checked whether maternal weight gain impacted the findings but there is also a DNA methylation signature associated with conditions such as prenatal hypertension (<https://doi.org/10.1161/HYPERTENSIONAHA.119.12634>). I'm not sure if there is overwhelming evidence that pregnancy complications are associated with birth order, but it seems

plausible that some of them may be.

How can we interpret the finding that the most significant CpG site (located in ACOT7) was not significant in children of African ancestry?

The authors should highlight the range of effect sizes in the discussion, not just for a single CpG (line 379). How big overall were the differences in DNA methylation associated with birth order? This would help the reader better understand the possible functional significance of the CpGs identified in this meta-analysis.

Smaller comments:

- Line 117, should rephrase to avoid referring to child as "it", maybe reword as "his/her family"
- Line 139-140, should rephrase to "the means and mechanisms by which these factors (related to birth order) impact child outcomes are not currently understood..."
- The last sentence of the introduction uses the term 'birth order' three times and could be reworded.
- I think Figure 1A should just be Figure 1 (line 250).
- On line 409-410, does the author mean that these traits have not previously been studied in relation to birth order, or they were studied and not found to be significantly related to birth order?

Reviewer #2 (Remarks to the Author):

In this manuscript, a collection of 16 birth cohorts from the PACE consortium is used to perform a meta-analysis to investigate the associations between neonatal DNA methylation in cord blood and birth order (defined as the number of deliveries a mother had at the time of a subject's birth). The results of the analysis show the presence of 341 CpGs that are differentially methylated in relation to birth order, some of which are associated to gene expression in the brain, the immune system, and the cardiovascular system. These findings provide further insight on the association between birth order and disease risk in later life.

Major comments:

- The study is of value because of the unique nature of the consortium data, although some limitations remains (such as limited data for some ethnicity groups), however the authors describe them in details in the conclusions.
- The distribution of birth order is very skewed, and very variable across cohorts. Not all the cohorts have the same range of birth order, and in general there seems to be a higher proportion of first borns. It is unclear to me whether the employed robust linear regression model (see also comment below) can provide unbiased estimates in this context. While modelling birth order as an ordinal variable is perfectly reasonable, I wonder if the results would replicate when modelling the birth order as a binary variable (first born vs. others).
- I believe the findings are nevertheless of value and of interest for the greater research community

Minor comments:

- In each cohort, the EWAS was implemented as a robust linear regression. Can the authors provide more details about the model, and explain the reason why this model was selected? For example, what assumptions of a traditional linear model were violated and how are these counteracted by the adoption of a robust method?

Reviewer #3 (Remarks to the Author):

Overall, this is a well-written manuscript. Key strengths included the use of multiple cohorts yielding a large sample size for this type of analysis. The sensitivity analysis probing cohort differences was a strength. The analysis generally appeared soundly completed. The main concerns limiting the potential impact of this paper include the need for rationale and clarity in the methods section, and a more targeted and clear discussion of the findings. Additionally, there needs to be some specific contribution of this study added to the introduction to really drive home the importance of doing a large EWAS with several different cohorts and the substantive contribution of the findings to the literature beyond a "catalog" that is very difficult to digest. Please find specific comments below.

Introduction:

- The introduction talks a lot about siblings, but the actual analysis was not done on siblings. Is there literature which compares birth order in unrelated children? As there is a review of specifically sibling comparison studies, it might be helpful to make sibling designs their own section with a heading and the current study in a different section with its own heading.

-

- Adding what gap this current study is filling in birth order/DNA methylation literature would be helpful to orient the reader.

Methods:

Label the supplemental file that has the DNA methylation extraction, preprocessing steps specifically. It is difficult to identify.

- On page 10 line 170/171: "If multiple participants within a sample set were from the same family, only one of them was randomly included in this study." First, it is unclear how many times/for how many families one sibling was included and how many other siblings were not included. Second, including only one child per family is potentially a missed opportunity. Leveraging siblings to examine differences in methylation controlling for many (not all) genetic and environmental confounds via a sibling comparison design would be a stronger test of the birth order hypothesis, and would be a valuable secondary or sensitivity analysis (assuming it could only be done on a subset).

- From the supplementary materials, it appeared as though QC was performed on each cohort first, and then again with the combined cohorts in the METAL analysis. Were any samples removed during the combined cohort QC?

- 14 cohorts used Illumina450K while 8 used EPIC chip, how might this have impacted the overlap in CpG sites? Which one did they use for EWAS? It may be helpful to report the number of CpG sites of each cohort since they ran EWAS in each of them.

- Birth order was ordinal, 1, 2, 3, etc. In the intro they mainly talked about the importance of being first born versus second, so does third or later birth order make a difference? From their demographic table it shows that the majority seem to be first borns.

- The use of cord blood needs to be clearly stated, they seem to only say "the blood".

- List the n of Latino participants as they did European and African even if the sample is smaller (page 13, line 236). Why was the sub-group analysis not performed for the Latino group? Also, were the African participants White or Black or both in the sub-group (since in Table 1, one was labeled Black African and the other just African). There could be very large differences between Black and White South African groups, and White African could actually be better represented by European ancestry.

- There were several instances of 'if X information was available'. One wonders how frequently and for

which study samples each piece of information is/is not available. More detail on each cohort is needed. The supplemental document has reasonable descriptions (though it doesn't mention selection factors, see below, or whether X information was available). I strongly suggest at least a summary of the commonalities and differences in the paper itself.

- Selection factor - please specify more clearly - again this is an 'if there was' --- what were the selection factors and for which cohorts did this apply? It is difficult to ascertain the quality of the meta-analysis without understanding the differences in cohorts.
- It is unclear why some (smoking) but not other (e.g., infection) prenatal maternal variables were included as covariates... reading further maternal weight gain was an additional covariate in a sensitivity analysis. The selection and treatment of maternal covariates is haphazard and not well described or justified.
- Please include the rationale for excluding CpG sites on sex chromosomes (e.g., rather than examining boys and girls separately and including them or another approach).
- In general, the analytic strategy is adequately described. However, it would be better to include rationale/reasons why each analysis was performed (e.g., what is to be gained from that specific analysis) especially since this analysis is so exploratory.

Results:

- Table 1: suggest at least including the range, preferably the range and % low birth weight. Including only measures of central tendency does not provide a very good understanding of the variability within each cohort
- Results should more clearly be labeled as exploratory.
- It may be useful to summarize the % positive and negative association (Table 2) given that there was sometimes quite a range of coefficient directions for each CpG site. What does it mean for the findings to have such variation?
- Throughout the results section, it is unclear whether hyper- or hypo- methylation in specific CpG sites and regions is associated with earlier or later born status – please specify the direction of effects throughout.
- African ancestry only yielded 1 out of 341 CpG sites, does this have to do with the sample size only being 378 compared to 7,484 of European ancestry? This, and/or possible mechanistic reasons, should be discussed.
- The plots of the sensitivity analysis are supposed to show that the sensitivity analysis and main analysis were similar, however the two graphs have different scales on the X and Y axis making them impossible to compare. They may need to be scaled to the same degree to actually look similar.

Discussion:

- The discussion was largely a restatement of some of the results. It was not clear what the findings mean, or how the specific results highlighted were chosen from the large supplemental tables to be focused on, nor how they all go together. The discussion read as a laundry list of things associated with birth order or CpG sites, with definitions of some biological processes interspersed, but without providing clear take-home messages or mechanistic hypotheses for how to understand the biological correlates of birth order in the context of human development. This paper would have a much higher impact should the authors provide one or two most likely candidate explanations or future directions, rather than a "catalog of candidates".
- The unrelated children (dropping siblings) limitation should be clearly stated in the methods section.
- "Cord blood of newborns" is introduced far too late (in the conclusion) and needs to be clearly stated in the methods.
- I disagree that using cord blood "helps to strengthen causal inference" because it does not rule out common causes to both methylation and distal outcomes. It can only rule out direct reverse causation.

Reviewers' comments:

Reviewer #1 (Remarks to the Author):

Thank you for the opportunity to review this interesting manuscript. The authors report the results of a multi-cohort meta-analysis showing over 300 CpG sites that are differentially methylated with regards to child birth order. I really enjoyed the clear presentation of (very complex) findings and the analytic choices and sensitivity analyses seemed carefully thought through and appropriate. I have a number of suggestions for strengthening the manuscript.

Comment: In the introduction, I thought too much space was dedicated to talking about postnatal explanations for birth-order differences, when this paper is focusing on prenatal explanations.

The second half of the first paragraph could be shortened, and more specific information could be provided in the second paragraph, to give more information about potential prenatal processes linking birth order to differential disease risk.

Response: *We agree that the extended description of the “hygiene hypothesis” was probably unnecessary, particularly since this is a debated topic outside of the scope of the current study. We therefore deleted that description in the first paragraph, along with another sentence that was somewhat superfluous.*

Comment: I also thought the study by Li et al about sibling pairs and twin births (line 141) could be better explained. Which siblings' DNA methylation were being compared? One of the twins and earlier/late-born siblings?

Response: *the Li, et al. study examined the correlation of singleton sibling DNA methylation before and after twin births, using HM450K array data genome-wide. Admittedly, this is a somewhat obscure analytical approach to embark upon, but the data yielded some interesting results. They measured blood DNA methylation in an Australian blood DNA methylation twin and family analysis (with a Korean family dataset used for replication) at adults (the blood DNA was obtained from middle age adults) and calculated the genome-wide average DNA methylation correlation between individuals who were 50% genetically identical to one another (being siblings) and found that DNA methylation correlation among siblings shifted to be more consistent (or correlated) following a twin pregnancy. These were indeed studies of multiple children from the same family – rather than unrelated subjects as used in the current study. We edited the description of this study in the Introduction to be clearer.*

Comment: The authors state that some studies included multiple children from the same family. How common was this? If there are enough cases, it would strengthen the authors' findings to investigate DNA methylation of birth-order related CpGs WITHIN families to see if the direction of associations found in between-family analyses plays out within-families. This was a limitation raised in the discussion. If the data do not exist to be able to address within-family differences

here, I think more should be said about this in the limitations. We know that family size is not a randomly distributed characteristic – it likely relates to parent and family characteristics that are also likely to be associated with disease risk (e.g., parental education).

Response: *DNA methylation studies of multiple children in the same family are somewhat rare, although there are multiple studies of twins in the literature. We agree that such studies are exciting and informative; however, we concentrate here on data made available by the PACE consortium which are nearly entirely studies of unrelated births - one each from a family. Yes we do note that family size varies drastically among populations (and within populations) and one of the most important take-aways of our study is the impact of changing family demographics leading towards a higher proportion of first-borns and so such “first-born DNA methylation patterns” having increasing relevance on public health.*

Comment: How prevalent were stillbirths in the sample? There are likely different physical effects associated with stillbirth vs live birth. Were sensitivity analyses conducted with and without these contributing to birth order (similar to how this was done for miscarriages and abortions?)

Response: *Stillbirth is typically defined as 20+ week birth of a deceased offspring - which is likely to have a stronger physiologic impact on maternal physiology than an early miscarriage – and that is why we decided to treat it as a live birth in analysis (some live births also occur close to 22 weeks). Among PACE studies, these birth outcomes were quantified in different ways (some birth certificate, some by questionnaire). Our own dataset (CCLS) defines stillbirth as termination of pregnancy after 20 weeks. However, it is extremely rare (0.6%), and excluding them is unlikely to affect final results.*

Comment: How was participant ancestry determined? Were there multiple ancestry children/families?

Response: *Ancestry was determined by self-report in most PACE studies; many of which were instituted within one race/ancestry group. A few studies with genome wide association data used principal components combined with self-report. There are indeed multiple ancestry families particularly in California where the predominant birth population are admixed Hispanics. Our approach at meta-analysis among world populations should select out CpG sites and genes that are associated with birth order across ancestral groups. A sensitivity analysis was included where we parsed out the European and African participants separately to compare.*

Comment: Can the authors clarify what the ‘selection factor’ (line 186) refers to?

Response: *We meta-analyzed results from a large number of childhood health and disease studies that were originally set up to study specific disease and adverse health conditions. In order to maximize power we included all healthy births available. The “selection factors” were postnatal disease characteristics which may be enriched within specific study*

populations. For instance, in our California study, this is a bivariate “case/control” selection factor – the cases are children who are healthy at birth but were diagnosed with acute leukemia later in childhood. Other studies used a nested case-control selection to augment cases of other conditions such as asthma or autism.

Comment: Birth order was treated as an ordinal variable in analyses. I think the results then reflect the assumption that each additional birth has an additive impact on DNA methylation. But, is there any evidence that there could be threshold effects? That is, are first-borns systematically different from later-borns, but later-borns do not differ from one another? Or, there is a difference up to 3 births and then no further differences? Could this be explored in these data?

Response: *Biologically speaking, both could be true: Methylation of some CpGs might be linearly associated with birth order, while others could have binary response (first born/any later born). Nevertheless, our model focuses on picking up the previous group, and they are potentially of greater biological interest (see next paragraph of this response). We did conduct additional analysis comparing first born versus others and results were similar. Specifically, CpGs overlapping ACOT7 were still the most statistically significant CpGs. We did not include this model in the final manuscript as we didn't think this added additional value to our findings. That said, in response to this comment, we have added information regarding this model briefly to the text.*

We agree that a more nuanced evaluation of each additional birth on DNA methylation would be interesting. If there were “threshold effects,” we would anticipate that such a threshold may occur after the first birth, since one would expect that a mother's first birth should provide the most permanent physiological changes. However, we note that subsequent births might have quantitatively additive effects –as an example, we show below an inset from a figure from Von Behren, et al, IJC 2011 which examined a large multi-state dataset on birth order and childhood cancer incidence. Note that, depending on the cancer type such threshold effects may vary – overall (all childhood cancers) it appears a clean additive effect, whereas for CNS cancers there is only an effect at more than two births. Our primary focus on disease is childhood cancers, and that is why we have chosen the current additive model.

Figure 1. Adjusted odds ratios and 95% confidence intervals for birth order categories by cancer type.

Comment: It was unclear from the description of analysis steps whether the DMR analysis was only conducted on the meta-analytic sample, or whether it was conducted separately by cohort and then meta-analyzed (as the DMPs were). Please clarify.

Response: The DMR analysis required only coefficients and *p* values from the meta-analysis of DMPs only, and therefore was not conducted individually on all the studies (which would have resulted in DMRs of different defined sizes and therefore difficult to combine). We added a clarification to the Methods to make this clear (p 42).

Comment: Were other pregnancy complications investigated? The authors checked whether maternal weight gain impacted the findings but there is also a DNA methylation signature associated with conditions such as prenatal hypertension (<https://doi.org/10.1161/HYPERTENSIONAHA.119.12634>). I'm not sure if there is overwhelming evidence that pregnancy complications are associated with birth order, but it seems plausible that some of them may be.

Response: We did not adjust for other pregnancy complications, as the quality or existence of such data varied widely among the datasets used for this analysis. We acknowledge that data on pregnancy complications may impact DNA methylation outcomes for some studies, and may exist for some datasets, but cannot assess this across studies used in this analysis.

Comment: How can we interpret the finding that the most significant CpG site (located in ACOT7) was not significant in children of African ancestry?

Response: The number of African subjects was small ($n=378$) and thus that analysis was underpowered compared to the European datasets, likely explaining lack of significance – however, the coefficients of association for this loci and others were consistent among datasets.

Comment: The authors should highlight the range of effect sizes in the discussion, not just for a single CpG (line 379). How big overall were the differences in DNA methylation associated with birth order? This would help the reader better understand the possible functional significance of the CpGs identified in this meta-analysis.

Response: *The range of effect sizes were two orders of magnitude among those reaching statistical significance (absolute value of effect sizes from 0.0001 to 0.01015) – in fact, many large effect size CpG sites did not rise high on the p-value due to the fact that they were less consistent in direction of effect across the large number of studies that were meta-analyzed. This represents the major power of the PACE meta-analysis approach – significant results need not have extreme effects as long as they are consistent. Subtle changes may be functionally significant if they alter epigenetic status at key gene regions that are tightly controlled. Based on the reviewer’s comment we examined the effect sizes and noticed that the largest significant effect sizes were did not include the most significant SNP. We have added a few sentences about CpG sites apart from the top hit about this in the Discussion (lines 431 and following).*

Comment: Smaller comments:

-Line 117, should rephrase to avoid referring to child as “it”, maybe reword as “his/her family”

Response: *Thank you for this suggestion. We have rephrased it to ‘their family’.*

Comment: -Line 139-140, should rephrase to “the means and mechanisms by which these factors (related to birth order) impact child outcomes are not currently understood...”

Response: *Thank you for the suggestion and we have changed the text as suggested.*

Comment: -The last sentence of the introduction uses the term ‘birth order’ three times and could be reworded.

Response: *Thank you for this suggestion. “Birth order” is a primary topic of the manuscript. This term is used once in the second last, and twice in the last sentence of the introduction. We think it is appropriate.*

Comment: -I think Figure 1A should just be Figure 1 (line 250). -On line 409-410, does the author mean that these traits have not previously been studied in relation to birth order, or they were studied and not found to be significantly related to birth order?

Response: *Thank you for pointing this out. We have changed the text as suggested.*

Comment: -On line 409-410, does the author mean that these traits have not previously been studied in relation to birth order, or they were studied and not found to be significantly related to birth order?

Response: Thank you for this comment. We meant that these traits were associated with birth order in our study, but were not studied/reported before in other studies to be associated with birth order. We have modified our text to make this clear.

Reviewer #2 (Remarks to the Author):

In this manuscript, a collection of 16 birth cohorts from the PACE consortium is used to perform a meta-analysis to investigate the associations between neonatal DNA methylation in cord blood and birth order (defined as the number of deliveries a mother had at the time of a subject's birth). The results of the analysis show the presence of 341 CpGs that are differentially methylated in relation to birth order, some of which are associated to gene expression in the brain, the immune system, and the cardiovascular system. These findings provide further insight on the association between birth order and disease risk in later life.

Major comments:

Comment: - The study is of value because of the unique nature of the consortium data, although some limitations remain (such as limited data for some ethnicity groups), however the authors describe them in detail in the conclusions.

Response: We thank the reviewer for positive comments on the value of these results. It is indeed a worthwhile opportunity to capture worldwide data using the consortium.

Comment: - The distribution of birth order is very skewed, and very variable across cohorts. Not all the cohorts have the same range of birth order, and in general there seems to be a higher proportion of first borns. It is unclear to me whether the employed robust linear regression model (see also comment below) can provide unbiased estimates in this context. While modelling birth order as an ordinal variable is perfectly reasonable, I wonder if the results would replicate when modelling the birth order as a binary variable (first born vs. others).

Response: this is a similar question as Reviewer 1. We considered a binary approach at first born/other. While there may be physiological reasons to consider a first birth as "set point" that prepares a mother's physiology for equivalent gestation for all subsequent births, we emphasize results for the ordinal model due to the successive effect on cancer risk shown above in the response to the first reviewer. This argues for an additional (additive) effect for each birth. However due to the interest in the binary output, which may become more important as families continue to have fewer children, we include a new supplementary table S10 with this data now referred to in line 323.

- I believe the findings are nevertheless of value and of interest for the greater research community

Response: thank you.

Minor comments:

Comment: - In each cohort, the EWAS was implemented as a robust linear regression. Can the authors provide more details about the model, and explain the reason why this model was selected? For example, what assumptions of a traditional linear model were violated and how are these counteracted by the adoption of a robust method?

Response: *We believe that robust linear regression is more suitable for DNA methylation EWAS models. When variables (for example, birth order in our study) are positively associated with DNA methylation, when birth order is higher, DNA methylation will also tend to have a higher variance. This violates the assumption of homoscedasticity in traditional linear regression models. Robust linear regression will give outliers less weight to correct for this bias. We also would like to point out that when there are no outliers in the dataset, robust linear regression still applies, because data points won't be re-assigned new weights, and in this case it works very similar to traditional linear regression.*

Reviewer #3 (Remarks to the Author):

Comment: Overall, this is a well-written manuscript. Key strengths included the use of multiple cohorts yielding a large sample size for this type of analysis. The sensitivity analysis probing cohort differences was a strength. The analysis generally appeared soundly completed. The main concerns limiting the potential impact of this paper include the need for rationale and clarity in the methods section, and a more targeted and clear discussion of the findings. Additionally, there needs to be some specific contribution of this study added to the introduction to really drive home the importance of doing a large EWAS with several different cohorts and the substantive contribution of the findings to the literature beyond a “catalog” that is very difficult to digest. Please find specific comments below.

Response: *thank you for the generally favorable comments on the manuscript. The point at our reaching a “catalog” of results as a main outcome was commented by other reviewers. We believe that all scientific literature is a step on the path to truth - and therefore the top “hits” (associations) in this manuscript are an approach to such truth. We can provide here a handbook (or catalog) of top hits - as virtually all such EWAS studies do – and these associations can be used by future researchers. We do not believe that any researcher can provide better or more valid results than we have provided here with regards to the variable “birth order” and we would wish our results can rest as a permanent (as all literature is) catalog of associations. We welcome any better nomenclature besides “catalog” as we sincerely believe that this is a great term for the list of CpGs and genes that we have identified.*

Introduction:

Comment: - The introduction talks a lot about siblings, but the actual analysis was not done on siblings. Is there literature which compares birth order in unrelated children? As there is a review of specifically sibling comparison studies, it might be helpful to make sibling designs their own section with a heading and the current study in a different section with its own heading.

Response: *This point was provided by the reviewer 1 - second comment. We respond to this point there along with changes to the manuscript.*

Comment: -- Adding what gap this current study is filling in birth order/DNAm literature would be helpful to orient the reader.

Response: *We are filling in an empty space – there is not a birth order-DNA methylation signature manuscript that we know of. Given disease risk associations with birth order, we explore the potential role of DNA methylation changes in mediating those risks. Also, the changing demographics worldwide and increased prevalence of “first borns” suggest this is potentially an important subject.*

Methods:

Comment: Label the supplemental file that has the DNA methylation extraction, preprocessing steps specifically. It is difficult to identify.

Response: *Thank you for this suggestion. We have modified our supplement file to have specific paragraphs dedicated to DNA methylation extraction and preprocessings.*

Comment: - On page 10 line 170/171: “If multiple participants within a sample set were from the same family, only one of them was randomly included in this study.” First, it is unclear how many times/for how many families one sibling was included and how many other siblings were not included. Second, including only one child per family is potentially a missed opportunity. Leveraging siblings to examine differences in methylation controlling for many (not all) genetic and environmental confounds via a sibling comparison design would be a stronger test of the birth order hypothesis, and would be a valuable secondary or sensitivity analysis (assuming it could only be done on a subset).

Response: *All of the consortium studies were set up with recruitment methods to collect sporadic, unrelated individuals. It was important therefore to maintain each subject as independent of all other study subjects to maintain statistical integrity. We therefore by necessity needed to leave out any family members as this violates independence. We agree in principle that a family-based analysis would be highly useful – allowing a conditional analysis which would gain power with smaller sample sizes. Upon inquiring our consortium colleagues we found out that no studies had inadvertently recruited family members, and therefore we could leave out this point in the Methods. Removal of any family members was part of our original instructions to the consortium members.*

Comment: - From the supplementary materials, it appeared as though QC was performed on each cohort first, and then again with the combined cohorts in the METAL analysis. Were any samples from removed during the combined cohort QC?

Response: QC was performed at the cohort level – as only summary statistics only (and no individual level data) were sent back to USC for the combined METAL meta-analysis, there was no additional opportunity for sample-level QC.

Comment: - 14 cohorts used Illumina450K while 8 used EPIC chip, how might this have impacted the overlap in CpG sites? Which one did they use for EWAS? It may be helpful to report the number of CpG sites of each cohort since they ran EWAS in each of them.

Response: In the methods used here, CpG loci that appear on both the 450K and EPIC arrays (N=XXXXXX) were meta-analyzed across both datasets (all 22 datasets) whereas CpG loci on one array or the other will only be meta-analyzed over the arrays that harbor data for such loci. For QC reasons some loci may be filtered out of some datasets even if they have the loci on the arrays. For example – in Table 2 top hit CpG loci, some “?” are displayed for some cohorts since data does not exist.

Comment: - Birth order was ordinal, 1, 2, 3, etc. In the intro they mainly talked about the importance of being first born versus second, so does third or later birth order make a difference? From their demographic table it shows that the majority seem to be first borns.

Response: This is a similar comment to the second reviewer, who wondered whether there may be threshold effects at certain birth orders. As noted in response to that question, we showed that some disease associations with birth order suggest an additive effect with each birth, justifying the additive ordinal model used here. We do however agree with the reviewer that most of the study's statistical power will result from comparison of first born to the second born since most families had at least those two siblings, dwindling in sample size for 3+ children. We do believe that later born children do make a difference physiologically since disease associations do show an additional effect.

Comment: - The use of cord blood needs to be clearly stated, they seem to only say “the blood”.

Response: We note that most of the studies used for this meta-analysis used cord blood, but some studies instead used neonatal blood from archived heel-prick blood spots. Certainly all of the studies have used blood from the child obtained neonatally. To clarify, we added this sentence in the first paragraph in the Methods: “All studies used neonatal blood – for most this was derived from the umbilical cord, and for some from heel-prick blood spots.” Detailed descriptions are included in the supplemental methods.

Comment: - List the n of Latino participants as they did European and African even if the sample is smaller (page 13, line 236). Why was the sub-group analysis not performed for the

Latino group? Also, were the African participants White or Black or both in the sub-group (since in Table 1, one was labeled Black African and the other just African). There could be very large differences between Black and White South African groups, and White African could actually be better represented by European ancestry.

Response: *All of the Latinos came from one study (California) and these were analyzed independently from non-Latino whites. The numbers of Latinos were shown in Table 1 – even though they came from one study, some were screened on 450K and others on EPIC as noted in the table. Table 1 studies are now bracketed more cleanly in the table which may help with clarity of presentation and any confusion about study origin. We agree that ancestral groups in South Africa are important to distinguish. Ancestry, and country of origin as a cultural identity, are independent factors. We present the data provided by the investigators of the study origin as they collected it with regards to self-reported identity (whether race or ethnicity). All analyses are performed stratified by major ethnic groups, and those race/ethnic groups with less than 15 total participants for analysis (at any one site) are excluded. We are not positioned to adjudicate the original study-specific race/ethnicity classifications and present the data as provided to us.*

Comment: - There were several instances of 'if X information was available'. One wonders how frequently and for which study samples each piece of information is/is not available. More detail on each cohort is needed. The supplemental document has reasonable descriptions (though it doesn't mention selection factors, see below, or whether X information was available). I strongly suggest at least a summary of the commonalities and differences in the paper itself.

Response: *We used the term “if available” if that particular study had collected the data in their original study instruments, for some of the less crucial covariates that were not part of the “main model.” We request the most comprehensive data possible but some studies do not have certain variables such as ‘prior miscarriage’. Readers can consult the Supplement to capture that information per study. As birth order is the primary descriptor and is consistent among all studies, all studies must include birth order and key (non-negotiable) covariates into the “main model” (birthweight, gestational age, race/ethnicity, etc) - if any of these key covariates are missing then the cohort was not included in the meta-analysis (or this manuscript).*

Comment: - Selection factor - please specify more clearly - again this is an 'if there was' --- what were the selection factors and for which cohorts did this apply? It is difficult to ascertain the quality of the meta-analysis without understanding the differences in cohorts.

Response: *In all cohorts, subjects were a collection of healthy children at birth. Some cohorts had selected subjects for DNA methylation analysis to study early developmental origins of diseases and therefore selected individuals who had contracted such diseases in a “nested case-control study within a birth cohort” design. If the samples subjected to DNA methylation analysis were selected in such a way, then this “selection factor” was included as a covariate to regress out any impact of this factor. For instance, the California study included future “childhood leukemia” as a selection factor. We clarified this with the statement in lines 216-223,*

“...selection factor (if there was selection on a phenotype to create the original DNA methylation dataset for each individual study – for instance leukemia status (case/control) in the CCLS study), maternal age (years), gestational age (weeks), birthweight (gram), and maternal smoking status (nonsmoker as 0, smoker as 1). Note that despite the “selection factor” all children were not identified as such at birth – any conditions or diseases selected were diagnosed/developed later in childhood.”

Comment: - It is unclear why some (smoking) but not other (e.g., infection) prenatal maternal variables were included as covariates... reading further maternal weight gain was an additional covariate in a sensitivity analysis. The selection and treatment of maternal covariates is haphazard and not well described or justified.

Response: Tobacco (smoking) exposure in pregnancy is well-known to produce DNA methylation impacts, and therefore was included in the “main model” as it was also universally collected among the participating cohorts. Maternal infection during pregnancy is information inconsistently collected by cohorts, and of differential quality (medical records-based versus self-report), and without demonstrated impact on neonatal DNA methylation profiles. We therefore elected to include maternal smoking in the main model. The section in Methods on sensitivity analyses is to evaluate the impact of certain covariates collected inconsistently among cohorts but potentially informative – maternal weight gain in this context is highlighted as it reflects in part nutritional status and potentially pregnancy complications – while either of these are even less consistently recorded at least “maternal weight gain” is a “hard” variable accessible to many cohorts. Its inclusion in the model did not materially change results providing some confidence in the birth order results.

Comment: - Please include the rationale for excluding CpG sites on sex chromosomes (e.g., rather than examining boys and girls separately and including them or another approach).

Response: We included sex as a covariate in all analyses, to regress out the very extensive DNA methylation impacts from sex on the autosomes which were the subject of a recent PACE meta-analysis [1]. We did not include X-chromosome CpGs as that would involve halving our sample size for the two separate sexes. The investigation of X-related CpGs is important and often excluded from such analyses as this. DNA methylation is a major mechanism in X-chromosome “Lyonization” which would severely complicate such an analysis - in females. We do not attempt it here but agree that methods to analyze birth order with this characteristic is of interest. We do not attempt this analysis here due to our original study design for all cohorts – such a reconsideration of this aspect would require an entirely new coordinated meta-analysis with all PACE members who participated in this current work. This is a wonderful idea and one that we will pursue, after completing the current study.

[1] Solomon O, Huen K, Yousefi P, et al. Meta-analysis of epigenome-wide association studies in newborns and children show widespread sex differences in blood DNA methylation. *Mutat Res Rev Mutat Res.* 2022;789:108415. doi:10.1016/j.mrrev.2022.108415

Comment: - In general, the analytic strategy is adequately described. However, it would be better to include rationale/reasons why each analysis was performed (e.g., what is to be gained from that specific analysis) especially since this analysis is so exploratory.

Response: *Thank you for this comment, and it points to the heart of our motivation to do this analysis. As childhood cancer researchers (that is the authors S. Li and J. Wiemels), we have long been interested in what biologic aspects are related to the consistent birth order associations among many different cancer sites. The potential for DNA methylation being a “reflection” of fetal developmental characteristics in tissues at birth is profound – and those characteristics that point to cancer risk will be extremely helpful for our etiologic studies in this area. We do not intend our analysis to be an exploratory analysis, as we have commandeered all the available studies worldwide to test associations with the PACE Consortium - this is an attempt to classify DNA methylation across multiple datasets worldwide with a descriptive variable that is universal - birth order. The use of multiple datasets and meta-analysis permits the derivation of results that are more robust than studies that involve one or two datasets only, so should be more definitive. Other PACE Consortium analyses have yielded state-of-the-science catalogs of DNA methylation sites and regions associated with birth characteristics and maternal pregnancy exposures (eg., birthweight, tobacco smoking). These PACE analyses involve a technology platform in-common among these studies (Illumina whole genome), an unprecedented opportunity to do such a study facilitated by the consortium.*

Each analysis described in the methods were explained as statistical models that include a minimal set of critical covariates - most of these are in-common with all PACE Consortium studies.

Results:

Comment: - Table 1: suggest at least including the range, preferably the range and % low birth weight. Including only measures of central tendency does not provide a very good understanding of the variability within each cohort

Response: *this is a good comment particularly as birthweight has a profound effect on DNA methylation, as exemplified in a prior PACE study that utilized most of the studies included in this analysis [1]. We have added this study as a reference to this point where readers could access more information about individual studies on this important covariate. All studies in PACE have mechanisms to exclude subjects whose birthweight (or other variables) are known to be incompatible with life. We did not specifically include instructions for each cohort to exclude*

[1] Küpers LK, Monnereau C, Sharp GC, et al. Meta-analysis of epigenome-wide association studies in neonates reveals widespread differential DNA methylation associated with birthweight. Nat Commun. 2019;10(1):1893. Published 2019 Apr 23. doi:10.1038/s41467-019-09671-3

Comment: - Results should more clearly be labeled as exploratory.

Response: The term “exploratory” is helpful when describing **proposed** studies that do not reach a power threshold (due to small sample size), or insufficient rationale. We explained the rationale for doing this study to address birth order in studies with available data around the world. The study has sample size, replication and validation status as a large meta analysis which we believe exceeds the standards of sample size criteria that would and will not necessarily label it as exploratory. However, the reviewer’s point is appreciated since none of these studies have originally been established to study birth order and DNA methylation. We added this statement to the first paragraph in Discussion, “As no single cohort was specifically designed to examine DNA methylation and birth order our results may be considered “exploratory” however the strength of the PACE Consortium allows confirmatory replication and validation”

Comment: - It may be useful to summarize the % positive and negative association (Table 2) given that there was sometimes quite a range of coefficient directions for each CpG site. What does it mean for the findings to have such variation?

Response: The presentation of “+” and “-” to be the summary - this is a typical presentation of rDNA methylation meta-analyses. As noted in the table footnote, the order is from largest to smallest cohort, so that more “weight” to judgment of systematic or true associations should be placed on initial or left side directions. This variation is likely due to systematic differences in individual studies that could involve variations in statistical precision, mQTL-ancestry differences, systematic error, or other cultural/dietary differences causing such variation. The meta-analytic methods account for this variation providing an estimate weighted by the statistical precision of the study specific estimates.

Comment: - Throughout the results section, it is unclear whether hyper- or hypo- methylation in specific CpG sites and regions is associated with earlier or later born status – please specify the direction of effects throughout.

Response: Consistent with the original DNA methylation model (line 224) results are presented as the direction of association that higher birth order and higher DNA methylation = “+” (hyper) methylation, and “-” = higher birth order and reduced methylation. To clarify this we added this sentence to Results (line 284): “In these and all data presented, positive coefficients refer to higher (hyper-) DNA methylation with later birth order compared to earlier, and negative coefficients refer to lower (hypo-) methylation with later birth order compared to earlier.”

Comment: - African ancestry only yielded 1 out of 341 CpG sites, does this have to do with the sample size only being 378 compared to 7,484 of European ancestry? This, and/or possible mechanistic reasons, should be discussed.

Response: Sample size is most certainly the main issue here, and we have added the phrase “likely due to small sample size.” (line 358) to account for this. There are likely additional reasons such as the influence of more complex ancestral haplotypes known to be important in

genetic studies, but we consider this to be beyond the scope of evaluation of the current dataset and the purview of this manuscript.

Comment: - The plots of the sensitivity analysis are supposed to show that the sensitivity analysis and main analysis were similar, however the two graphs have different scales on the X and Y axis making them impossible to compare. They may need to be scaled to the same degree to actually look similar.

Response: *Those two plots are not meant to be compared to each other; instead each plot has its own comparison of effect sizes with and without the inclusion of the sensitivity variable (different for each inset figure). Still, a reader might be curious if there were differences in the impact of those two variables which could be qualitatively evaluated with comparison of graphs with the same axes scales. We updated the figure as such, and note that there are slight differences in shape but not likely with any meaningful consequence.*

Discussion:

Comment: - The discussion was largely a restatement of some of the results. It was not clear what the findings mean, or how the specific results highlighted were chosen from the large supplemental tables to be focused on, nor how they all go together. The discussion read as a laundry list of things associated with birth order or CpG sites, with definitions of some biological processes interspersed, but without providing clear take-home messages or mechanistic hypotheses for how to understand the biological correlates of birth order in the context of human development. This paper would have a much higher impact should the authors provide one or two most likely candidate explanations or future directions, rather than a "catalog of candidates".

Response: *We agree with the reviewer about the context of the discussion, which is generally an attempt to ascribe biological meaning to the statistical associations listed in Results. Some of these results are restated for the purpose of then conjecturing on the biology via the known function of the gene. The manuscript is a statistical exploration strength in numbers of cross-sectional observations of human children at birth, with strength in the number of participants and cohorts studied, but we have not performed any mechanistic experiments around the observations noted. Like most of the scientific literature, this is an incremental advance in expanding our collective knowledge base. It is important to include longer lists in the supplement as a permanent record of gene loci that could be related to birth order, as future research groups will study aspects of birth order and other birth characteristics on DNA methylation and postnatal outcomes with likely further gain knowledge and replication from their own investigations of these genes in relation to health effects.*

Comment: - The unrelated children (dropping siblings) limitation should be clearly stated in the methods section.

Response: *This is noted in the "Definition of birth variables" section of Methods. (line 199)*

Comment: - "Cord blood of newborns" is introduced far too late (in the conclusion) and needs to be clearly stated in the methods.

Response: *We have added in Methods that neonatal blood was used for all studies from either cord blood or heel-prick blood spots (line 184-186)*

Comment: - I disagree that using cord blood "helps to strengthen causal inference" because it does not rule out common causes to both methylation and distal outcomes. It can only rule out direct reverse causation.

Response: *We have removed the statement about causal inference from the discussion.*

Reviewers' comments:

Reviewer #1 (Remarks to the Author):

While the revised version of the manuscript is definitely improved, several of my concerns were not adequately addressed and require further attention.

There is still too much space devoted to postnatal explanations for birth-order differences in the first paragraph. Almost half of the first paragraph is devoted to postnatal explanations, even though as the authors say, this is not a focus of the current analysis. I would delete all of this information (lines 160-167 in tracked manuscript) and instead elaborate more on the statement that "some disease trends may be related to this changing demographic" – what changes in disease trends have been noted that could be due to birth order? An example would be helpful here instead of the detail about postnatal explanations. The information about the postnatal explanations should be moved to the discussion and could be mentioned only briefly in the introduction (i.e., "While postnatal experiences including child rearing practices and infection exposure may also differ based on birth order, they are not likely to be related to the pre-birth environment and are not a subject of the current analysis")

The first sentence of the second paragraph in the introduction could also be re-worded: "Importantly, first-borns experience different gestational environments than their later-born siblings, as indexed using a variety of different biomarkers. These environments may impact later disease risk..." (line 168-170).

I did not understand the authors' rebuttal regarding cohorts with multiple children from the same families. In their response to my comment, the authors state that PACE cohorts are "nearly entirely studies of unrelated births". In their response to another reviewer who made the same comment, they mention that there were no related individuals excluded and that this detail could be left out of the methods. Yet the text was not removed from the methods. So which is it – were individuals from related births removed or not? If not, this detail should be removed.

Information about how ancestry was determined should be added to the text since it was important to the interpretation of models and sensitivity analyses.

The limitation regarding unrelated individuals instead of same-family siblings (like 509-511) should be expanded to mention that family size is not a random factor and could be associated with parent and family characteristics such as parental education and income, pertinent factors that could also be expected to be associated with child disease risk. The current explanation that this "introduced noise and decreased the reliability of our findings" is a bit vague.

Reviewer #2 (Remarks to the Author):

The authors have addressed my comments and I believe the manuscript is now in a suitable form for publication.

Reviewer #3 (Remarks to the Author):

The current paper compared DNA methylation values of neonates from several international samples to test the impact of birth order on methylation values across CpG sites. Strengths of the manuscript remain, include the analytically sound meta-analysis of multiple cohorts. Authors responded to a few points – adding some minor clarifications in the paper. Although in many cases responses to reviewer

comments were reasonable in the rebuttal letter, it was disappointing to see that relatively little of the information requested by reviewers made it into the revised manuscript itself – and so in reading the revised MS many of the same concerns remain.

For example, it remains unclear in the introduction that the current study did not use sibling comparison designs in the introduction and to provide information about the sample size of related individuals in the study (two separate comments in the original review). The authors pointed to their response to Reviewer #1's question about the Li et al., sibling study in the rebuttal letter. However, this comment did not clarify or relate the specific comments we provided. At a minimum, the introduction should clearly state that the analysis is including singletons only (and not siblings) around line 202 (Here we aimed...). The authors also say in the rebuttal in response to the sample size of related individuals/how many were dropped: "Upon inquiring our consortium colleagues we found out that no studies had inadvertently recruited family members, and therefore we could leave out this point in the Methods." – then why is there a sentence in the paper saying "if multiple participants within a sample set were from the same family, only one of them was randomly included in this study"? It begs the question of 'how often did this occur', and if the answer is 'never' then that information can easily be added, or that sentence removed and/or replaced with something more relevant and less question-raising. The sentence "Removal of any family members was part of our original instructions to the consortium members." From the rebuttal seems at odds with this though.

Additionally, there were many occasions where the authors gave explanations in their comment that they neglected to add into the paper to also inform possible future readers. For example, when we asked for them to add their contribution to the literature to orient the reader, and the gap that this paper fills, they gave an acceptable explanation but did not add this to the paper.

Lastly, and most importantly, we asked for a description of the substantive contributions of the findings to the field and a clearer, applied explanation of the findings was not addressed by the authors. This is the main drawback of this paper, because as results are currently interpreted in the discussion, it is difficult to digest. The discussion section specifically may negatively impact the likelihood of being cited. While this paper contributes research to a field of work that would benefit from these findings, the authors did not take this opportunity to clarify the importance of the results and how they can be applicable to child health or development in a broader sense related to the CpG sites findings rather than continuously listing genes that may be impacted.

Reviewers' comments:

Reviewer #1 (Remarks to the Author):

While the revised version of the manuscript is definitely improved, several of my concerns were not adequately addressed and require further attention.

Comment: There is still too much space devoted to postnatal explanations for birth-order differences in the first paragraph. Almost half of the first paragraph is devoted to postnatal explanations, even though as the authors say, this is not a focus of the current analysis. I would delete all of this information (lines 160-167 in tracked manuscript) and instead elaborate more on the statement that “some disease trends may be related to this changing demographic” – what changes in disease trends have been noted that could be due to birth order? An example would be helpful here instead of the detail about postnatal explanations. The information about the postnatal explanations should be moved to the discussion and could be mentioned only briefly in the introduction (i.e., “While postnatal experiences including child rearing practices and infection exposure may also differ based on birth order, they are not likely to be related to the pre-birth environment and are not a subject of the current analysis”)

Response: *We appreciate this comment that we should focus our Introduction on relevant “first born” phenomena that are related to the pregnancy and neonate, rather than postnatal exposures. We have removed the section suggested by the reviewer, and now included a couple of new references that review the increasing incidence of relevant diseases that have been linked to birth order. We removed the statement about postnatal exposures; instead going immediately to biological observations of “first born” that may be related to DNA methylation changes – notably, nutritional and hormonal differences and the limited number of DNA methylation observations in the literature. The suggestion for the first sentence of that paragraph by the reviewer is quite good.*

Postnatal explanations for birth order-related disease effects have dominated the past literature – which is appropriate given the strong impact of such exposures, particularly the impact of infections brought home by older children to the later born. We moved this discussion to the Discussion as suggested by the reviewer – please check lines 502-510 in the resubmitted manuscript.

Comment: The first sentence of the second paragraph in the introduction could also be re-worded: “Importantly, first-borns experience different gestational environments than their later-born siblings, as indexed using a variety of different biomarkers. These environments may impact later disease risk...” (line 168-170).

Response: *As noted above, we appreciate the suggestion for the transitional sentence to the second paragraph, and used it verbatim, line 128.*

Comment: I did not understand the authors' rebuttal regarding cohorts with multiple children from the same families. In their response to my comment, the authors state that PACE cohorts are “nearly entirely studies of unrelated births”. In their response to another reviewer who made the same comment, they mention that there were no related individuals excluded and that this detail could be left out of the methods. Yet the text was not removed from the methods. So which is it – were individuals from related births removed or not? If not, this detail should be removed. Information about how ancestry was determined should be added to the text since it was important to the interpretation of models and sensitivity analyses.

As this is an association study (as most EWAS are), we instructed all participating PACE studies to include one family member only – specifically, if multiple participants (2 or more) family members were included in their original study cohort, only one can be included in this study (per original instructions). We now have looked back at the response letter from last review, and our original instructions to all PACE cohorts. We did say in the response letter that “nearly entirely” unrelated births were included – this is a false statement as absolutely all births included must be unrelated to others in the cohort, as this was an original study plan for all cohort participants. Our Methods clearly state this in line 185-8 of the current manuscript (this line was included in previous manuscript versions) and we also added the clarification “...to maintain independence of all study subjects.” We apologize for adding the unclear statement to the previous response, which was not truthful to our original design, and thank the reviewer for persisting on this point – it is critical and related to best practices for an EWAS meta-analysis as we present here.

With regards to ancestry – we did not attempt to account for genetic ancestry in this study. In our prior study considering genetic ancestry, we found that accounting for local SNP (mQTL) effects had a subtle but not substantial impact on DNA methylation from birth characteristics (in that study, birthweight: Clin Epigenetics, 2022 PMID: 36457128). In the current study, most cohorts did not have SNP scans on their study subjects making ancestry evaluation impossible given the wide international study subject capture. The DNA

methylation impact from birth is here a fundamental trait randomized by race/ethnicity – in this case our top hits for DNA methylation.

Comment: The limitation regarding unrelated individuals instead of same-family siblings (like 509-511) should be expanded to mention that family size is not a random factor and could be associated with parent and family characteristics such as parental education and income, pertinent factors that could also be expected to be associated with child disease risk. The current explanation that this “introduced noise and decreased the reliability of our findings” is a bit vague.

Response: Indeed, family size is not necessarily a random factor among study participants –family size is determined in part by cultural characteristics which vary by country, SES status, and other factors that may also impact DNA methylation status and disease risk. That said, birth order itself is a shared ordinal characteristic in every family – no matter the genetic or cultural origin. We are able to statistically control for some characteristics at the individual level among all study sites: sex, cell type distributions, maternal age, gestational age, birthweight, and maternal tobacco use – however not SES or its components such as income and education. We have explained these issues more thoroughly in the Discussion, now appearing in lines 477-83.

Reviewer #2 (Remarks to the Author):

Comment: The authors have addressed my comments and I believe the manuscript is now in a suitable form for publication.

Response: Thank you for the confidence in the presentation, and trust that the new additional changes have further improved the manuscript.

Reviewer #3 (Remarks to the Author):

Comment: The current paper compared DNA methylation values of neonates from several international samples to test the impact of birth order on methylation values across CpG sites. Strengths of the manuscript remain, include the analytically sound meta-analysis of multiple cohorts. Authors responded to a few points – adding some minor clarifications in the paper. Although in many cases responses to reviewer comments were reasonable in the rebuttal letter, it was disappointing to see that relatively little of the information requested by reviewers made it into the revised manuscript itself – and so in reading the revised MS many of the same concerns remain.

Response: Thank you for persisting on this issue and we went back to the prior reviewer response to find comments and responses to which we did not change the manuscript. We've made a number of additional changes as noted below.

a. removal of postnatal factors attributed to birth order in the introduction (also brought up by reviewer 1 in the second review)

b. Addition of a comment about stillbirth sensitivity analysis in Results (line 270)

c. Removed the repetitive “birth order” in the last two sentences of Introduction as suggested by reviewer 1 in the last review.

d. Added that this is the first time that a birth order/EWAS study was performed, line 162, to help bolster the novelty of this study and utility for further investigations on birth-order related diseases

e. added that no cohort data was removed at the meta-analysis stage (all cohorts included, line 268)

Comment: For example, it remains unclear in the introduction that the current study did not use sibling comparison designs in the introduction and to provide information about the sample size of related individuals in the study (two separate comments in the original review). The authors pointed to their response to Reviewer #1's question about the Li et al., sibling study in the rebuttal letter. However, this comment did not clarify or relate the specific comments we provided. At a minimum, the introduction should clearly state that the analysis is including singletons only (and not siblings) around line 202 (Here we aimed...). The authors also say in the rebuttal in response to the sample size of related individuals/how many were dropped: “Upon inquiring our consortium colleagues we found out that no studies had inadvertently recruited family members, and therefore we could leave out this point in the Methods.” – then why is there a sentence in the paper saying “if multiple participants within a sample set were from the same family, only one of them was randomly included in this study”? It begs the question of ‘how often did this occur’, and if the answer is ‘never’ then that information can easily be added, or that sentence removed and/or replaced with something more

relevant and less question-raising. The sentence “Removal of any family members was part of our original instructions to the consortium members.” From the rebuttal seems at odds with this though.

Response: *We appreciate this point, which was shared by Reviewer #1 – and in response above we noted that our prior reviewer response included a statement which was not true (contradicting the other statement noted by this reviewer that family members must be removed if found in a cohort). Family-based study designs are useful but cannot be mixed with association studies such as this one that depend on independence of study subjects. We apologize for the misrepresentation in the prior rebuttal letter.*

Comment: Additionally, there were many occasions where the authors gave explanations in their comment that they neglected to add into the paper to also inform possible future readers. For example, when we asked for them to add their contribution to the literature to orient the reader, and the gap that this paper fills, they gave an acceptable explanation but did not add this to the paper.

Response: *Thank you for this point and we worked more diligently to justify our contribution to the literature as explained in the response to the comment just below this one. Looking back to the prior review we see that the reviewer wanted – we also looked back on our prior review response and noted above (in the first comment, Reviewer #3) some additional changes to the manuscript.*

Comment: Lastly, and most importantly, we asked for a description of the substantive contributions of the findings to the field and a clearer, applied explanation of the findings was not addressed by the authors. This is the main drawback of this paper, because as results are currently interpreted in the discussion, it is difficult to digest. The discussion section specifically may negatively impact the likelihood of being cited. While this paper contributes research to a field of work that would benefit from these findings, the authors did not take this opportunity to clarify the importance of the results and how they can be applicable to child health or development in a broader sense related to the CpG sites findings rather than continuously listing genes that may be impacted.

Response: *Thank you for this point, and this manuscript cannot be helpful unless it is read and understood by the scientific community, and the data utilized for further study which will elicit citations. Yes, as the reviewer noted we’ve listed a number of genes and speculated on their meaning in the prior discussion, without a wider context of their potential use. We believe that this gene listing will provide tools to investigate specific diseases that are linked to birth order – as such an investigator interested in type-1 diabetes, high blood pressure, allergy, leukemia or any other disease linked to birth order, and has DNA methylation data can pull the associated CpG sites presented in this paper to investigate as mediators between birth order (or other risk factors) and their diseases of interest. Most investigations focused on birth order have addressed postnatal exposures only – here we direct attention on prenatal developmental processes that should also be impactful and can set the stage for further investigation of a wide variety of diseases related to birth order.*

To respond to this comment we made changes in the Introduction and Discussion. We added lines in the Introduction to introduce the idea that the findings in this manuscript have implications in the study of the many diseases related to birth order, when DNA methylation data is available (lines 128 and following). In response to reviewer 1 we focused the intro more on prenatal origins of birth order effects, therefore pointing the finger away from postnatal impacts which dominate the prior literature. In the Discussion we have spelled out DNA methylation as worthy of study as a potential mediator of effects of birth order on disease risk – and our list of DNA methylation associations presented here can serve as a roadmap for such studies (lines 373-78). In addition, we emphasize the current study investigation on prenatal environments with those postnatal (such as infections) on lines 502-510.